# *Dyrk1a* gene dosage in glutamatergic neurons has key effects in cognitive deficits observed in mouse models of MRD7 and Down syndrome

Véronique Brault[1], Thu Lan Nguyen[1], Javier Flores-Gutiérrez[1], Giovanni Iacono[2], Marie-Christine Birling[3], Valérie Lalanne[3], Hamid Meziane[3], Antigoni Manousopoulou[4¤], Guillaume Pavlovic[3], Loïc Lindner[3], Mohammed Selloum[1], Tania Sorg[1], Eugene Yu[5,6], Spiros D. Garbis[4¤], Yann Hérault[1,3]*

1 Université de Strasbourg, CNRS, INSERM, Institut de Génétique et de Biologie Moléculaire et Cellulaire, IGBMC, Illkirch, France, 2 Department of Molecular Biology, Radboud Institute for Molecular Life Sciences, Radboud University, Nijmegen, the Netherlands, 3 Université de Strasbourg, CNRS, INSERM, CELPHEDIA, PHENOMIN, Institut Clinique de la Souris, Illkirch, France, 4 Institute for Life Sciences, University of Southampton, School of Medicine, Southampton, United Kingdom, 5 The Children's Guild Foundation Down Syndrome Research Program, Genetics and Genomics Program and Department of Cancer Genetics and Genomics, Roswell Park Comprehensive Cancer Center, Buffalo, New York, United States of America, 6 Genetics, Genomics and Bioinformatics Program, State University of New York At Buffalo, Buffalo, New York, United States of America

¤ Current address: Proteas Bioanalytics Inc., BioLabs at The Lundquist Institute, Torrance, California, United States of America

* herault@igbmc.fr

**Data Availability Statement:** The data underlying the results presented here in the study are available

## Abstract

Perturbation of the excitation/inhibition (E/I) balance leads to neurodevelopmental diseases including to autism spectrum disorders, intellectual disability, and epilepsy. Loss-of-function mutations in the *DYRK1A* gene, located on human chromosome 21 (Hsa21,) lead to an intellectual disability syndrome associated with microcephaly, epilepsy, and autistic troubles. Overexpression of DYRK1A, on the other hand, has been linked with learning and memory defects observed in people with Down syndrome (DS). *Dyrk1a* is expressed in both glutamatergic and GABAergic neurons, but its impact on each neuronal population has not yet been elucidated. Here we investigated the impact of *Dyrk1a* gene copy number variation in glutamatergic neurons using a conditional knockout allele of *Dyrk1a* crossed with the Tg (Camk2-Cre)4Gsc transgenic mouse. We explored this genetic modification in homozygotes, heterozygotes and combined with the Dp(16*Lipi-Zbtb21*)1Yey trisomic mouse model to unravel the consequence of *Dyrk1a* dosage from 0 to 3, to understand its role in normal physiology, and in MRD7 and DS. Overall, *Dyrk1a* dosage in postnatal glutamatergic neurons did not impact locomotor activity, working memory or epileptic susceptibility, but revealed that *Dyrk1a* is involved in long-term explicit memory. Molecular analyses pointed at a deregulation of transcriptional activity through immediate early genes and a role of DYRK1A at the glutamatergic post-synapse by deregulating and interacting with key post-synaptic proteins implicated in mechanism leading to long-term enhanced synaptic

upon the Zenodo repository with the DOI: 10.5281/zenodo.5121262.

**Funding:** This work has been supported by the National Centre for Scientific Research (CNRS), the French National Institute of Health and Medical Research (INSERM), the University of Strasbourg (Unistra), the French state funds through the "Agence Nationale de la Recherche" under the frame programme Investissements d'Avenir [ANR-10-IDEX-0002-02, ANR-10-LABX-0030-INRT, ANR-10-INBS-07 PHENOMIN, ANR-18-CE16-0020 DYRKDOWN to YH]. This project has received funding from the Jérôme Lejeune foundation and the European Union's Horizon 2020 research and innovation programme under grant agreement No 848077. The funders had no role in study design, data collection and analysis, decision to publish, or preparation of the manuscript.

plasticity. Altogether, our work gives important information to understand the action of DYRK1A inhibitors and have a better therapeutic approach.

## Author summary

The Dual Specificity Tyrosine Phosphorylation Regulated Kinase 1A, DYRK1A, drives cognitive alterations with increased dose in Down syndrome (DS) or with reduced dose in DYRK1A-related intellectual disability syndromes (ORPHA:268261; ORPHA:464311) also known as mental retardation, autosomal dominant disease 7 (MRD7; OMIM #614104). Here we report that specific and complete loss of *Dyrk1a* in glutamatergic neurons induced a range of specific cognitive phenotypes and alter the expression of genes involved in neurotransmission in the hippocampus. We further explored the consequences of *Dyrk1a* dosage in glutamatergic neurons on the cognitive phenotypes observed respectively in MRD7 and DS mouse models and we found specific roles in long-term explicit memory with no impact on motor activity, short-term working memory, and susceptibility to epilepsy. Then we demonstrated that DYRK1A is a component of the glutamatergic post-synapse and interacts with several component such as NR2B and PSD95. Altogether our work describes a new role of DYRK1A at the glutamatergic synapse that must be considered to understand the consequence of treatment targeting DYRK1A in disease.

## Introduction

Down syndrome (DS; Trisomy 21), is the leading genetic cause of mental retardation. Among genes present on the Hsa21, the Dual-specificity Tyrosine-(Y)-phosphorylation-Regulated Kinase 1A (*DYRK1A*), the mammalian homologue of the *Drosophila* minibrain (*mnb*) gene that is essential for normal neurogenesis [1,2], is a target for improvement of DS cognition [3]. In addition, 21q22.13–22.2 microdeletion syndrome associated to DYRK1A (ORPHA:268261) and intellectual deficit due to loss-of-function mutations in *DYRK1A* [4,5] (ORPHA:464311; also known as mental retardation, autosomal dominant disease 7: MRD7; OMIM #614104) show neurodevelopmental anomalies [4,6–10], making this gene a critical dosage-sensitive gene for cognitive phenotypes. MRD7 is characterized by severe intellectual disability (ID), speech and motor delay, autism spectrum disorder (ASD), epileptic seizures and microcephaly [1,2,6,10–14]. The rodent *Dyrk1a* is expressed in foetal and adult brains in dividing neuronal progenitors and later in the adult cerebellum, olfactory bulb and hippocampus [15–18]. DYRK1A is a serine/threonine kinase with many substrates and interactors involved in cell proliferation, neuronal morphogenesis, synaptogenesis and synaptic function [19]. Roles of DYRK1A have been revealed in brain development and neuronal differentiation via the control of critical signalling pathways such as AKT, MAPK/ERK and STAT3 or in synaptic function via the NFAT pathway [20,21]. Transgenic mice with either excess or haploinsufficiency of *Dyrk1a* show cognitive deficits like those observed in patients with specific impairment of hippocampal-dependent learning and memory [22–24].

Among the mechanisms proposed to underlie the cognitive deficits in DS is glutamatergic and GABAergic neurotransmitter dysfunction. Studies of the DS mouse model Ts65Dn trisomic for about 56% of the human chromosome 21 (Hsa21) syntenic region on mouse chromosome 16 (Mmu16) have revealed excess GABAergic input leading to reduced activation of

NMDA receptors and reduction of long-term potentiation (LTP) in the hippocampal CA1 and dentate gyrus (DG) areas [25]. In addition, enhanced hippocampal long-term depression (LTD) has also been observed in the hippocampi of Ts65Dn mice in response to sustained activation of excitatory synapses and attributed to excessive signalling via NMDA receptors [26–28]. Recent evidence supports the contribution of DYRK1A to changes in glutamatergic neurotransmission, with a BAC transgenic mouse line overexpressing *Dyrk1a*, showing alterations in glutamatergic synaptic proteins and normalization of *Dyrk1a* in Ts65Dn mice improving synaptic plasticity, GABAergic/glutamatergic balance, learning and memory [29,30]. *Dyrk1a* heterozygous knockout mice also present a reduction in the dendritic arborisation and the spine density of glutamatergic pyramidal neurons of the cerebral cortex and alterations in glutamatergic and GABAergic synaptic proteins [24,31,32]

In this context, we hypothesized that change in the dosage of *Dyrk1a* in glutamatergic neurons of the hippocampus and cortex of DS and MRD7 mouse models somehow alter their development and/or normal working in adult brain, leading to the cognitive deficits observed in DS or *Dyrk1a* haploinsufficiency models. Analysis of DYRK1A function in glutamatergic neurons using a knockout approach is not possible as its full KO is homozygote lethal [32]. Thus, we decided to change the gene dosage of *Dyrk1a* in glutamatergic neurons either in a disomic (inactivation of one or two copies of *Dyrk1a*) or trisomic (going back to two copies of *Dyrk1a*) context. We selected the Tg(Camk2a-Cre) transgene to target the Cre recombinase in postnatal glutamatergic neurons within the forebrain [33] and we used the Dp(16)1Yey trisomic mouse model (abbreviated as Dp1Yey) containing a segmental duplication of the 22.9 Mb *Lipi-Zfp295* region including *Dyrk1a*, to return to two copies of *Dyrk1a* in the glutamatergic neurons of the Dp1Yey. This model has the advantages to include 65% of Hsa21 mouse gene orthologs and to be devoid of the 50 DS-irrelevant trisomic genes that are present on the Ts65Dn mini chromosome [34]. Dp1Yey mice present defects in working memory, long-term episodic memory, and associative learning. In addition to those tests, we also tested the impact of *Dyrk1a* gene dosage on the mouse social behaviour as MRD7 persons display autistic traits.

## Results

### *Dyrk1a* is expressed in Camk2a-positive cells and its full inactivation in the glutamatergic neurons induces brain defects

*Dyrk1a* is ubiquitously expressed in different neuronal cell populations of the brain but with regional differences: the protein level being higher in the olfactory bulb, cerebellar cortex, cortical structures and granular and pyramidal cell layers of the hippocampus [35]. We checked DYRK1A expression in adult glutamatergic neurons, by co-immunohistochemical localisation with an antibody against CAMK2A and DYRK1A. In the wild-type adult mouse, both proteins were found in pyramidal and granular neurons of the hippocampus and dentate gyrus and in neurons of the cortex (S1 Fig).

We wanted to better understand the function of DYRK1A in adult glutamatergic neurons, circumventing the effect of Dyrk1a on early brain development. For this, we inactivated both copies of *Dyrk1a* using a conditional approach, to generate a full knock-out in those neurons. A floxed *Dyrk1a* allele (*Dyrk1a*^cKO allele) was designed such that exon 7 that codes for the serine/threonine protein kinase active site signature domain was flanked by two *loxP* sites (Fig 1A). We used the Tg(Camk2aCre)4Gsc transgene [33] to generate the *Dyrk1a*^Camk2aCre allele (shortened as *Dyrk1a*^C). This transgene starts to express the Cre recombinase at three weeks of age in the cortex and hippocampus, when most neural circuits are already formed, as described previously [33,36,37]. Our laboratory has previously shown that Cre deletion occurs in around 60% of the Camk2Cre-positive cells in the cortex and in around 80% of the Camk2Cre-positive

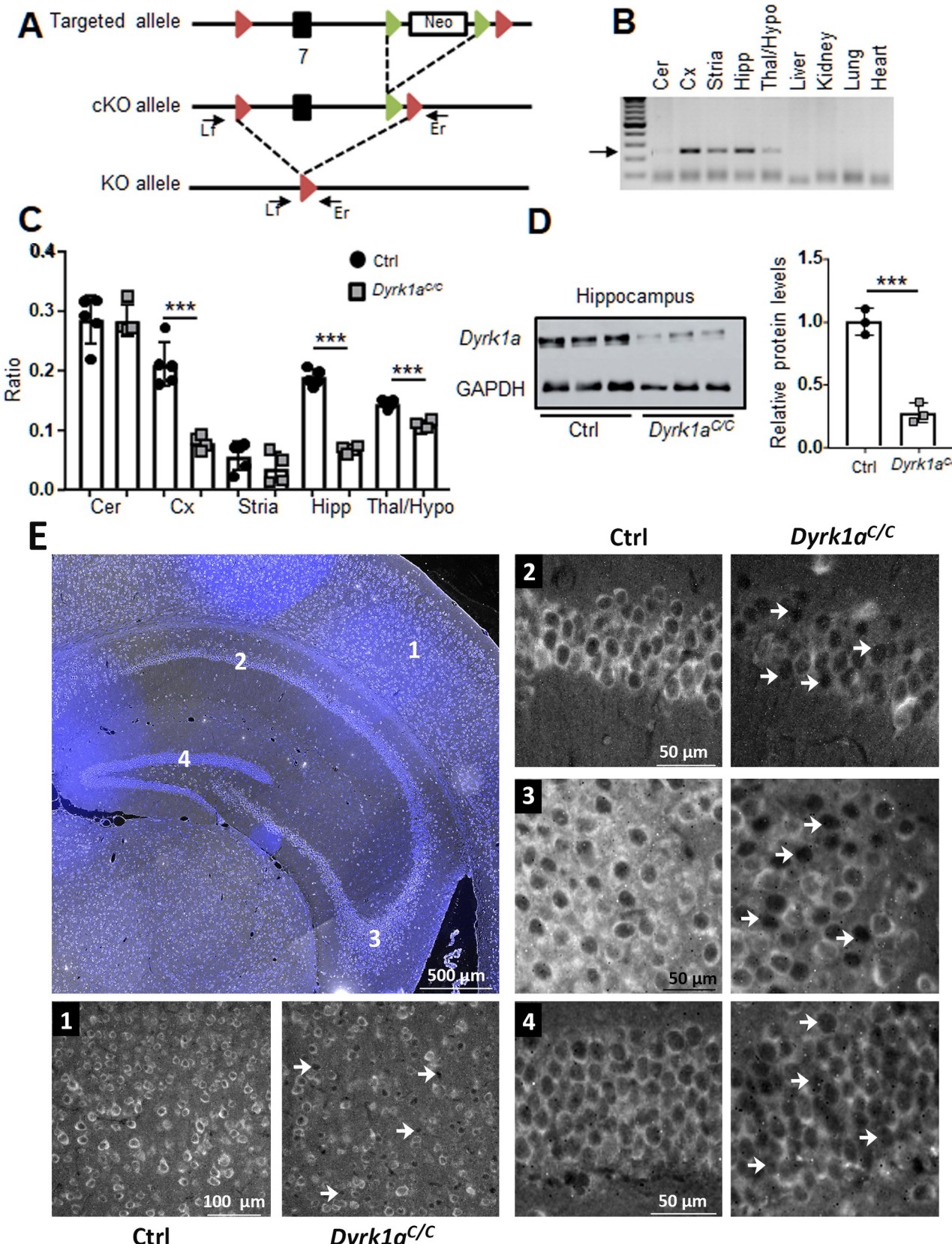

**Fig 1. Generation of mice deficient for Dyrk1a in the glutamatergic neurons.** (A) Targeting strategy for conditional inactivation of *Dyrk1a*. Exon 7 containing the serine/threonine protein kinase active site was flanked with loxP sites (red arrowheads) in two steps: a targeted allele was first generated by homologous recombination in ES cells, then *in vivo* expression of the Flp recombinase resulted in recombinaison of the FRT sites (green arrowheads) and removal of the selection cassette (white box) generating the conditional allele (cKO). The knock-out allele (KO) was observed in the brain of *Dyrk1a$^{C/C}$*. Arrows represent primers for PCR genotyping. (B) Genomic DNA was isolated from different organs from a *Dyrk1a$^{C/C}$* mouse and genotyped for the presence of the knock-out allele with primers Lf and Er, giving a 232 bp PCR product for the KO allele. (C) Ratio of relative mRNA of *Dyrk1a* in different brain structures in *Dyrk1a$^{C/C}$* and disomic control mice. (D) Autoradiographic image and quantification of immunoblots of DYRK1A protein in the hippocampus of *Dyrk1a$^{C/C}$* mice relative to control mice. Band intensities were estimated using ImageJ and normalized against the loading control GAPDH. Data are presented as point plots with mean ± SD with unpaired Student's t-test, *p<0.05, **p<0.01, ***p<0.001 (n = 5 ctrl and 4 *Dyrk1a$^{C/C}$* hippocampus for mRNA analysis and n = 3 per genotype for protein analysis). (E) DYRK1A immunohistochemistry of coronal brain sections around Bregma -1.5 mm (Paxinos adult mouse brain atlas, Franklin and Paxinos, 1997) at the level of the hippocampus and of the cortex from a control and a *Dyrk1a$^{C/C}$* mouse (2-month-old males). Expression of DYRK1A was detected in the cytoplasm of the neurons of the control (Ctrl) and a net decrease of expression was observed in *Dyrk1a$^{C/C}$* neurons as visible in the high-magnification images of the cortex (1), Cornus Ammonis 1 (CA1, 2), Cornus Ammonis 3(CA3, 3) and Dentate Gyrus (DG, 4) (white arrows point at neurons with little or no DYRK1A signal). DYRK1A signal is in grey while the blue color corresponds to DAPI in the representative image of the analyzed brain region. DAPI signal was removed from high-magnification images to better visualize DYRK1A signal and nuclei hence appear as black dots on those images.

cells of the hippocampus [38]. We checked the ability of the Cre to recombine the *Dyrk1a* floxed allele in *Dyrk1a$^{Camk2aCre/Camk2aCre}$* (recombination of both *Dyrk1a$^{cKO}$* alleles with the Cre recombinase; noted here *Dyrk1a$^{C/C}$*) mice. The generation of the deleted allele was detected by PCR analysis exclusively in brain areas where *Camk2a* is expressed (Fig 1B). Quantification of *Dyrk1a* mRNA in different brain regions confirmed that *Dyrk1a* is expressed at different relative levels in brain subregions (Fig 1C). Nevertheless, decrease of the *Dyrk1a* transcripts was found in the hippocampus, cortex and thalamus/hypophysis but not in the cerebellum of *Dyrk1a$^{C/C}$* mice (Fig 1C). Loss of the DYRK1A protein was confirmed in the hippocampus by Western blot analysis (Fig 1D) and immunohistology (Fig 1E). This reduction was more evident within the pyramidal cell layers of the CA1 and CA3 composed mostly of glutamatergic neurons.

We analysed the implication of DYRK1A in glutamatergic neurons by looking at brain morphology and cognitive phenotypes. Brain weight was significantly decreased in *Dyrk1a$^{C/C}$* mice compared to control mice (90% of the control weight; Fig 2A). We selected sections at Bregma -1.5 to have the hippocampus and the cortex for morphometric analysis, as those two structures are known to be impacted in by *Dyrk1a* gene dosage change [31,39] (Fig 2B). Contouring of the brain unravelled reduced surface area of the total brain surface in *Dyrk1a$^{C/C}$* mice (~88% of control; Fig 2C). The area of the hippocampus including the cornus ammonis fields (CA1, CA2 and CA3) and dentate gyrus (DG) did not significantly differ between the two genotypes (Fig 2D). We measured the thickness of the oriens layer at the CA1, CA2 and CA3 levels, of the pyramidal layer at the CA1 level, of the radiatum layer, of the CA1 and DG molecular layers and of the granular layer of the DG and did not find any difference between control and *Dyrk1a$^{C/C}$* mice (S2A Fig). Within the hippocampus, the dorsal CA1 region has a central role in episodic-like memory formation and was shown to consolidate memory during object novelty discrimination [40–42]. We counted cells within a specific frame in the middle of the CA1, but found no difference in cell density (S2B Fig). Dyrk1a was reported to increase cell density in layer VI of the somatosensory cortex [39]. Specific decrease in cortical thickness was observed at the level of the dorsal motor cortex (~78% of controls, Fig 2E) and of the somatosensory cortex (~76% of controls, Fig 2E) whereas decrease in thickness at the more ventral auditory cortex level was not significant (Fig 2E). Measurements of the thickness of different layers in the somatosensory cortex (Fig 2F) indicate decrease in the thickness of molecular layer I, external granular and pyramidal layers II/III, internal pyramidal layer V and internal polymorphic layer VI (Fig 2G). Only the internal granular layer IV was found unchanged (Fig 2G). To investigate how change in cellularity might relate to cortical thickness, we counted the number of cells present in SSC layers II-III, V and VI. We found that cellular

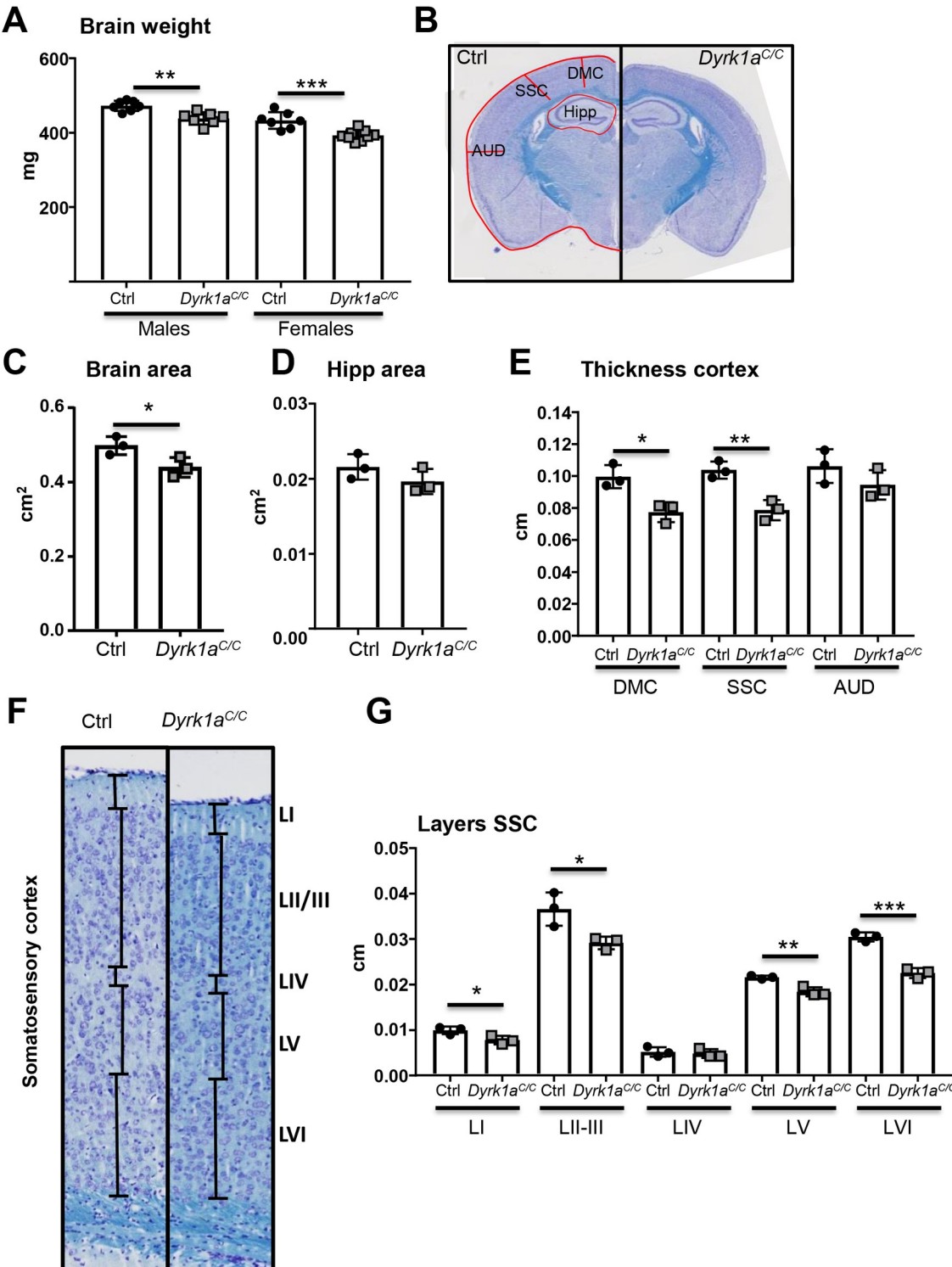

**Fig 2. Consequence of *Dyrk1a* inactivation in glutamatergic neurons on brain morphology.** (A) Brain weight from male and female mice aged 3 months old (n = 7–9 per genotype). (B) Representative coronal sections of control (Ctrl) (left) and *Dyrk1a*$^{C/C}$ (right) brains at Bregma -1.5 stained with cresyl violet and luxol blue that were used for measurements (Magnification 20X). (C) Dot plots of total brain area measurements (red line around the brain in B). (D) Dot plots of hippocampal areas (red area around hipp in B). (E) Measurements of the thickness of the cortex at the 3 levels represented by red lines in figure B. (F) Representative cresyl violet and luxol blue stained coronal sections of somatosensory cortex layers in control (Ctrl) and *Dyrk1a*$^{C/C}$ brains at Bregma -1.5. (G)

Measurements of the thickness of the different layers presented in F. Data are presented as point plots with mean ± SD with unpaired Student's t-test, *p<0.05, **p<0.01, ***p<0.001 (n = 3 females per genotype). AUD: auditory cortex, SSC: somatosensory cortex, DMC: dorso motor cortex, Hipp: hippocampus.

density in layers II-III, V and VI was increased by about 30% in *Dyrk1a*$^{C/C}$ mice (Fig 3D). We confirmed that neuronal density was indeed increased in the SSC of *Dyrk1a*$^{C/C}$ mice by labelling neurons with the NeuN marker and counting NeuN-positive cells (Fig 3B and 3E). We used S100b immunostaining to visualize astrocytes in the cortex to see if decreased cortical size could be due to decrease number of this cell population (Fig 3C and 3F). As for the total cell population and the neuronal population, we found the same number of S100b-positive cells in both genotypes resulting in an increased density in the cortex of *Dyrk1a*$^{C/C}$ mice (Fig 3F).

## Full *Dyrk1a* inactivation in the glutamatergic neurons impacts general behaviour and cognition

To analyse mouse behaviour in *Dyrk1a*$^{C/C}$ mice, we first focused our attention on locomotor activity and exploratory activity. Measurement of horizontal, or vertical, locomotor activity during circadian cycle did not differ in *Dyrk1a*$^{C/C}$ mice compared to control mice (S3A and S3B Fig). The analysis of exploratory behaviour in a novel environment (open field (OF) test) indicated normal locomotor activity for the *Dyrk1a*$^{C/C}$ mice (S3C Fig) but their exploratory pattern was altered as they spent significantly more time in the centre of the OF (Fig 4A), suggesting a decreased anxiety. This phenotype was confirmed in the elevated plus maze with *Dyrk1a*$^{C/C}$ mice spending significantly more time in the open arms than control mice (Fig 4B). In this test, *Dyrk1a*$^{C/C}$ mice were also more active, visiting more arms than the control mice (Fig 4C). The locomotor performance was assessed in the rotarod task. *Dyrk1a*$^{C/C}$ mice exhibited slightly better performance in this test than their control littermates, with an increase in latency to fall, indicating that motor balance is not affected in those mice (Fig 4D).

Impact of the loss of *Dyrk1a* in glutamatergic neurons on cognition was evaluated using different memory tests. Working memory was assessed by recording spontaneous alternation in the Y-maze. The percentage of alternation between the three arms was similar between *Dyrk1a*$^{C/C}$ and control mice (S3D Fig) indicating a normal working memory in both genotypes. In this test, the number of visited arms during the 5 min session was not significantly different in the *Dyrk1a*$^{C/C}$ mice compared to the controls, although those mice showed more variability (S3E Fig). Long-term explicit memory requiring the hippocampus and related medial and temporal lobe structures was tested with the novel object recognition test (NOR) with 24 hours delay. Although *Dyrk1a*$^{C/C}$ mice showed as much interest exploring the objects during the presentation session (Fig 4E) and during the discrimination session (Fig 4F) compared to control animals, they did not make any difference between the two objects during the retention trial (Fig 4G) by contrast to control mice who spent significantly more time on the novel object compared to the familiar one. Thus, the NOR test unravelled a deficit in long-term explicit memory in the *Dyrk1a*$^{C/C}$ mice. Then, we tested associative learning using the contextual fear-conditioning test. During the habituation, mice showed the same basal level of freezing whatever their genotypes were (Fig 4H, Habituation). However, *Dyrk1a*$^{C/C}$ mice showed significantly less freezing than control mice during contextual discrimination, indicating poorer performance in contextual learning (Fig 4H, Context). During the cued learning, *Dyrk1a*$^{C/C}$ mice responded like control mice to the conditioned stimulus, indicating normal cued fear (S3F Fig). As decreased freezing could be due to a deficit in pain sensitivity rather than a deficit in memory, we tested the mice in the hot plate test. *Dyrk1a*$^{C/C}$ mice had a decreased latency to elicit a first response to noxious thermal stimulus, suggesting that they were more sensitive to

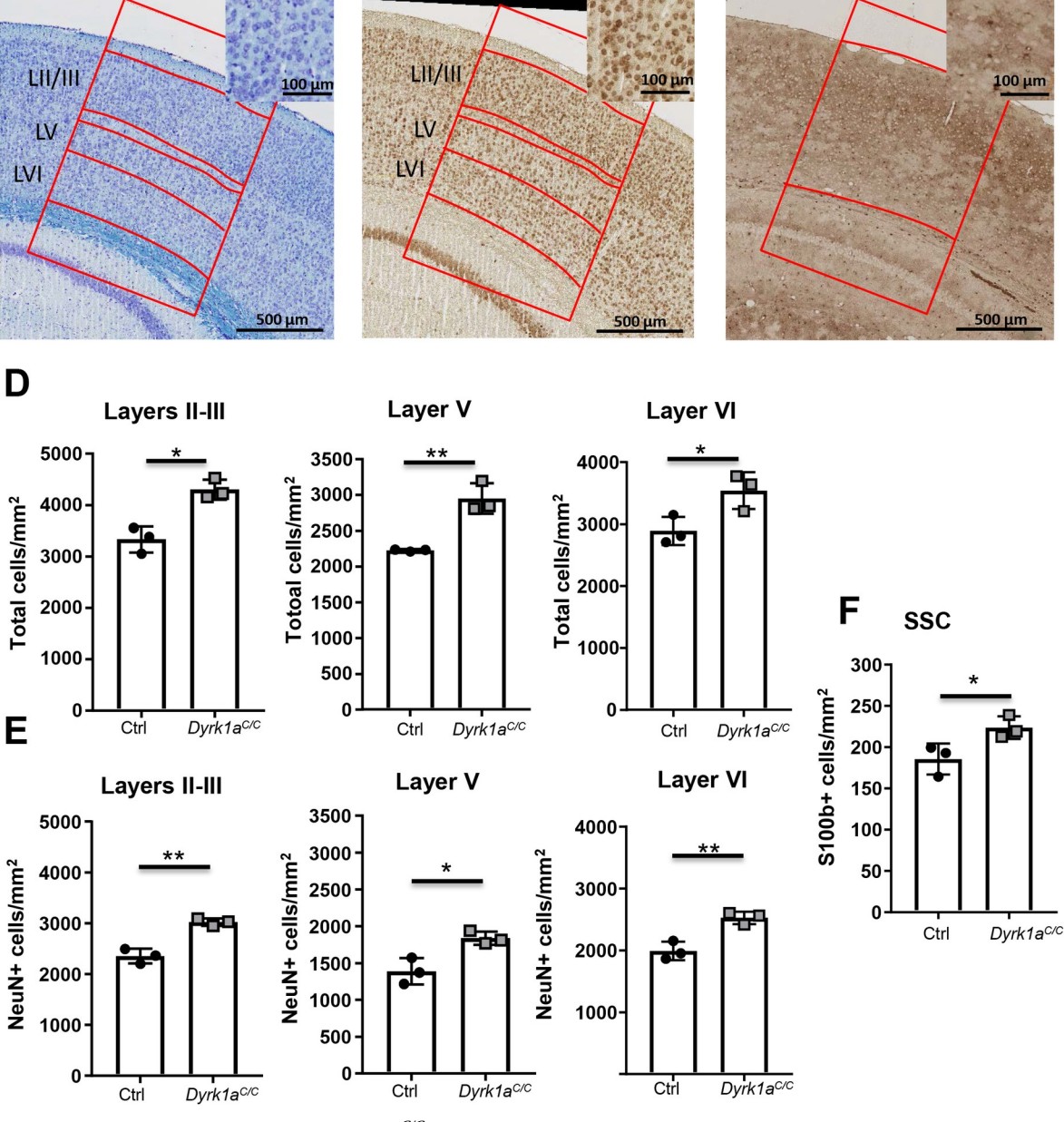

**Fig 3. Analysis of cell density in the SSC of *Dyrk1a^{C/C}* mice.** Representative cresyl violet and luxol blue stained (A), NeuN-labeled (B) and S100b-labeled (C) coronal sections at Bregma -1.5 with the counting frames. (D-F) Relative density of cells counted within a frame of 0.1 cm width at the level of the SSC. Data are presented as point plots with mean ± SD with unpaired Student's t-test, *p<0.05, **p<0.01, ***p<0.001 (n = 3 females aged 3 months per genotype).

pain than control mice (Fig 4I). As pain sensitivity was never tested in *Dyrk1a* knock-out heterozygous mice (shortened as *Dyrk1a^{+/-}*), we also tested those mice in the hot plate. We also found that those mice are more sensitive to pain (Fig 4J).

As in Human *DYRK1A* heterozygous mutations lead to autistic behaviour in MRD7, mouse sociability was investigated in this full inactivation of *Dyrk1a* in the glutamatergic neurons. We presented an empty cage and a cage containing a congener to the tested mouse and

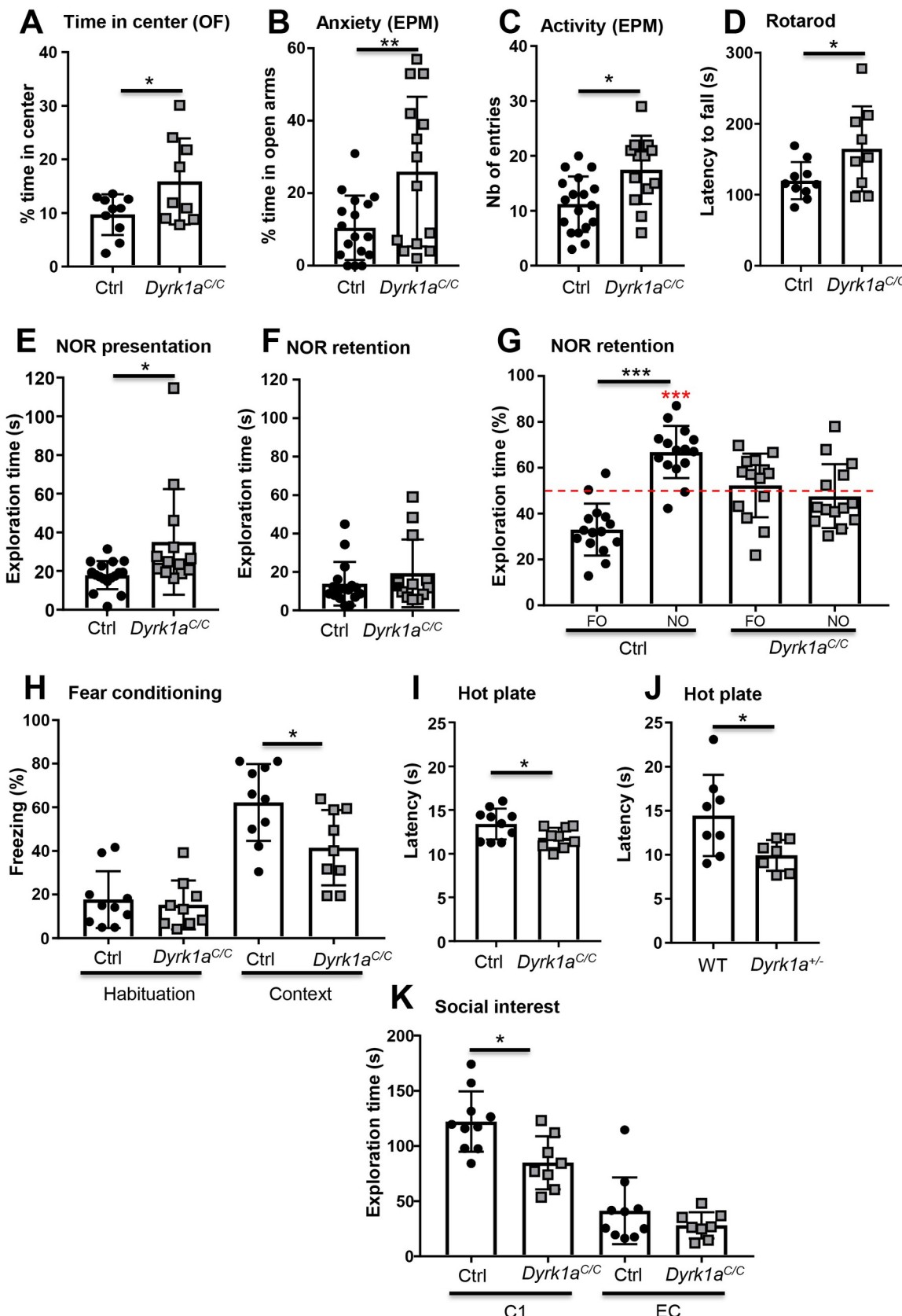

**Fig 4. Impact of *Dyrk1a* inactivation in glutamatergic neurons on general behavior, locomotor activity and cognition.** (A) The exploratory behavior of a new environment was analyzed by the percentage of time spent in the center of an open field over 30 min of test. *Dyrk1a*$^{C/C}$ mice spent more time in the center of the arena, suggesting that they are less anxious. (B) Confirmation of the phenotype in a new group of mice using the Elevated Plus Maze test (EPM) with *Dyrk1a*$^{C/C}$ mice showing a higher percentage of time spent in the open arms of the maze. (C) Mouse activity, measured by the number of entries in both open and closed arms, was increased in *Dyrk1a*$^{C/C}$ animals. (D) Evaluation of the locomotor performance on the rotarod during consecutive trials with increased rotational speeds. The latency is the mean of 3 independent trials. *Dyrk1a*$^{C/C}$ mice showed increased performance. (E) In the NOR test, the percentage of time spent exploring the familiar (FO) and the new (NO) objects show that control mice spend significantly more time on the NO while *Dyrk1a*$^{C/C}$ mice do not make any difference between the two objects. (F) Exploration times of the two identical objects during the presentation phase of the NOR show that *Dyrk1a*$^{C/C}$ mice tend to have an increased exploration time. (G) Total exploratory time of the two objects during the test indicates that the absence of object discrimination of the *Dyrk1a*$^{C/C}$ mice is not due to a lack of interest of the objects. (H) Percentage of freezing time during the habituation phase (before the foot shock; basal level of activity) (Habituation; Mann-Whitney rank sum test, p = 0.71) and during the 6 min of contextual exposure 24 hours later indicate a deficit of contextual learning in *Dyrk1a*$^{C/C}$ mice. (I-J) Pain sensitivity was evaluated by measuring the mouse latency to elicit a response to pain when put on a plate (52°C). In this test, *Dyrk1a*$^{C/C}$ and *Dyrk1a*$^{+/-}$ mice had a lower threshold than their control littermates. (K) Interest in social interaction was measured by the time spent sniffing the cage containing a congener (C1) during the Crawley test. This time was reduced for *Dyrk1a*$^{C/C}$ mice compared to control mice (unpaired t-test p = 0.01) while the time spent exploring the empty cage (EC) was the same between the two groups (unpaired t-test p = 0.23). Data are presented as point plots with mean ± SD. Statistical analyses were done with unpaired Student's t-test or Mann-Whitney rank sum test if normality test failed, except E: paired T-test FO vs NO and one sample T-test vs 50% mean (in red: ***p<0.001); A, D, H, I and K: tests were done on young males (1.5–3.5 months old depending on the test with animals aged ± 3 weeks), n = 8–10 per genotype,. B-C (EPM) and E-G (NOR) were done with another batch of males a 4 months of age, n = 18 and 16 controls for each test respectively and 13 *Dyrk1a*$^{C/C}$ (2 animals were removed from the NOR test because they did not reach the minimum exploration time during the presentation session); *p<0.05, **p<0.01, ***p<0.001.

measured the time spent by the tested mouse to sniff either cage. Both *Dyrk1a*$^{C/C}$ and control mice showed social preference as they spent significantly more time sniffing the cage containing the congener than the empty cage (S3G Fig, Social preference). However, the total amount of time spent with their congener was decreased in *Dyrk1a*$^{C/C}$ mice compared to control mice whereas the time spent exploring the empty cage did not differ (Fig 4K). Preference for social novelty was tested by placing a new congener in the empty cage. Both genotypes spent significantly more time sniffing the cage containing the new congener compared to the cage with the familiar one (>60% of the time allocated for the new congener) (S3H Fig, Social novelty preference). There was also no significant difference in the total time control and transgenic animals spent sniffing both congeners (S3I Fig, Social contact).

Finally, as *DYRK1A* haploinsufficiency in human is causing epilepsy, we challenged the homozygous inactivation in *Dyrk1a*$^{C/C}$ and control mice with two different doses of the seizure-provoking agent pentylenetetrazol (PTZ) and the occurrence of myoclonic, clonic and tonic seizures was scored. At both 30 mg/kg (S3J Fig) and 50 mg/kg (S3K Fig), *Dyrk1a*$^{C/C}$ susceptibility to seizure was like control mice. Altogether, those results indicate that *Dyrk1a* full inactivation in glutamatergic neurons does not increase susceptibility to PTZ-induced seizure.

Hence, *Dyrk1a* inactivation in glutamatergic neurons only impacts specific cognitive function such as explicit long-term memory, contextual fear memory and exploratory behavior while having no impact in others such as working memory, social behaviour and epileptic susceptibility.

## *Dyrk1a* inactivation in the glutamatergic neurons lowers expression of genes involved in neurotransmission in the hippocampus, while enhancing expression of genes implicated in the regulation of transcription

The hippocampus is a key structure in memory formation. Long-term object recognition memory analysed in the NOR test was shown to require interaction between the hippocampus and the perirhinal cortex [43–45] while contextual fear memory involves a neural circuit including the hippocampus, amygdala and medial prefrontal cortex [46]. Although we could not detect any morphological defect in the hippocampus of *Dyrk1a*$^{C/C}$ mice, those mice are

defective in both long-term recognition and contextual fear memories. To unravel the potential molecular mechanisms underlying the learning defects of *Dyrk1a*$^{C/C}$ mice, we performed genome-wide transcriptional profiling (RNA-seq) of *Dyrk1a*$^{C/C}$ and control mice in the hippocampus at postnatal day 30, as the brain is fully developed and mature at this stage. Analysis of the RNA-seq exon reads was performed using hypergeometric test and Bonferroni correction (DEseq algorithm, P<0.025) as previously published [38,47,48] and identified 297 up-regulated and 257 down-regulated genes in *Dyrk1a*$^{C/C}$ compared with controls (S1 Table). To determine the putative cell types associated with the deregulated genes, we compared the sets of up- and down-regulated genes with the markers of hippocampal cell types obtained from single cell RNA-seq (see Materials and Methods). As a result, up-regulated genes were enriched in oligodendrocyte-expressed genes (hypergeometric test, *bonferroni* corrected, P<3.4E-13) whereas down-regulated genes were enriched in neuronal markers (hypergeometric test, *bonferroni* corrected, pyramidal markers P<2.3E-4, interneuronal markers P<8.3E-3) (S2 Table). This decrease in neuronal markers expression is not reflected by a decrease in neuronal cells in the hippocampus as we did not observe a decrease in the thickness of the pyramidal cell layers or the DG granular cell layer, and did not find a deficit in cell density within the CA1 of the *Dyrk1a*$^{C/C}$ hippocampus (S2 Fig). Next, we performed GO enrichment analyses of the lists of up- and down-regulated genes with using a *Benjamini* cut-off of P < 0.05 (Table 1). The strongest enrichments for up-regulated genes were related to transcriptional regulation and DNA methylation. This category of genes did not have any overlap with the oligodendrocyte overexpressed genes at the exception of the SRY-related HMG-box transcription factor *Sox8* and we did not find any enriched specific function for the list of the oligodendrocyte markers that are up-regulated in *Dyrk1a*$^{C/C}$ hippocampi. We counted the number of Olig2 + cells in the corpus-callosum and found no difference between *Dyrk1a*$^{C/C}$ and control mice, suggesting that increased oligodendrocyte markers is not due to an increased number of oligodendrocytes (S4 Fig). Using the list of up-regulated genes that are oligodendrocyte-specific, we searched if this list was enriched for specific biological processes with FunRich

**Table 1. Gene ontology enrichment derived from up- and down-regulated genes in the *Dyrk1a*$^{C/C}$ hippocampus.**

| Up-regulated genes | | | | | |
|---|---|---|---|---|---|
| GO TERM | BENJ. PVAL | COUNTS | ENRICHMENT | EXPECTED | DESCRIPTION |
| GO:0006351 | 7.31e-05 | 54/1882 | 21.68 | 2.40 | Transcription, DNA-template |
| GO: 0006357 | 5.04e-05 | 51/1805 | 20.79 | 2.45 | Regulation of transcription from RNA polymerase |
| GO:0044212 | 9.82e-05 | 32/847 | 9.76 | 3.28 | Transcription regulatory region DNA binding |
| GO:0006325 | 3.12e-03 | 26/614 | 7.07 | 3.68 | Chromatin organization |
| GO:0009653 | 1.65e-03 | 52/2069 | 23.86 | 2.18 | Anatomical structure morphogenesis |
| **Down-regulated genes** | | | | | |
| GO TERM | BENJ. PVAL | COUNTS | ENRICHMENT | EXPECTED | DESCRIPTION |
| GO:0045202 | 3.09e-08 | 36/905 | 8.99 | 4.01 | Synapse |
| GO:0043005 | 1.10e-05 | 39/1285 | 12.76 | 3.06 | Neuron projection |
| GO:0098793 | 4.23e-04 | 18/354 | 3.51 | 5.12 | Presynapse |
| GO:0070382 | 1.02e-03 | 13/185 | 1.84 | 7.08 | Exocytic vesicle |
| GO:0008021 | 2.64e-03 | 12/167 | 1.66 | 7.24 | Synaptic vesicle |
| GO:0070044 | 1.04e-03 | 4/5 | 0.05 | 80.58 | Synaptobrevin 2-SNAP-25-syntaxib-1a complex |
| GO:0031201 | 1.96e-01 | 6/49 | 0.49 | 12.33 | SNARE complex |
| GO:0001505 | 1.13e-02 | 12/191 | 1.90 | 6.33 | Regulation of neurotransmitter levels |
| GO:0061025 | 6.36e-02 | 10/151 | 1.50 | 6.67 | Membrane fusion |
| GO:0051648 | 6.78e-02 | 11/188 | 1.87 | 5.89 | Vesicle localisation |

(http://funrich.org) and found that eight of the twenty-seven listed genes were implicated in cell communication and signal transduction, suggesting that *Dyrk1a* inactivation in glutamatergic neurons might have a secondary impact on communication between those neurons and oligodendrocytes. Interestingly, among up-regulated genes, we found *Nr4a1 (Nurr77)*, *Arc (Arg3.1)*, *Npas4*, *Fos (cFos)*, *Egr1 (Zif268) and Fosb*, six immediate-early genes (IEGs) encoding proteins involved in transduction signals that are induced in response to a wide variety of cellular stimuli and that are implicated in neuronal plasticity. Looking at known late response genes known to be activated by NPAS4 [49] in glutamatergic neurons, only three out of thirty-four (10%) of them were significantly deregulated in the hippocampus of *Dyrk1a*$^{C/C}$ compared to control mice (S4 Table), with *Fam198b* being up-regulated and *Csrnp1*and *Slc2a1* being down-regulated. Among target genes of NPAS4 shared between excitatory and inhibitory cells, four out of twelve (~30%) that we looked at were found deregulated in the hippocampus of *Dyrk1a*$^{C/C}$ compared to control mice (*Lmo2* and *Fosl2*, up-regulated; *Mylk* and *Nptx2*, down-regulated). Down-regulated genes found in the hippocampus transcriptome of *Dyrk1a*$^{C/C}$ mice were associated with presynaptic vesicle exocytosis, regulation of neurotransmitter levels and neuron projection, and pointed at a perturbation of chemical synaptic transmission via the deregulation of proteins involved in synaptic vesicle exocytosis. Particularly, genes coding for proteins of the SNARE complex (*Snap25*, *Stx1a*, *Napa* and *Napb*), regulating its activity (*Doc2b*, *Snph*) or implicated in vesicular synaptic cycle (*Anxa7*, *Amph*, *Syn2*, *Syngr1*) were found downregulated in the hippocampus of *Dyrk1a*$^{C/C}$ mice. This complex is known to mediate synaptic vesicle docking and fusion with the presynaptic membrane during neuromediator release. The SNARE complex was recently found, also with NPAS4, as a common pathway misregulated in models of DS overexpressing DYRK1A [50].

## Behavioural defects are induced in *Dyrk1a*$^{C/+}$ mice while partial rescue of memory alterations is observed in Dp1Yey/*Dyrk1a*$^{C/+}$ mice

We investigated the respective consequences of *Dyrk1a* dosage in glutamatergic neurons on the cognitive phenotypes observed respectively in MRD7 and DS mouse models. We analysed mice heterozygous for the *Dyrk1a* knockout allele in glutamatergic neurons to investigate the implication of this gene in the cognitive phenotypes of MRD7. We also performed a rescue experiment consisting on the return to two copies of *Dyrk1a* in the glutamatergic neurons of Dp1Yey trisomic mice. For this, we compared animals carrying *Dyrk1a*$^{Camk2aCre/+}$ (noted *Dyrk1a*$^{C/+}$), Dp1Yey, and Dp1Yey/*Dyrk1a*$^{Camk2aCre/+}$ (noted Dp1Yey/*Dyrk1a*$^{C/+}$), with *Dyrk1a*$^{cKO/+}$ as controls in behavioural tests. Mouse locomotor behaviour was tested in the open-field (OF). *Dyrk1a*$^{C/+}$ mice did not show any significant difference in locomotor activity compared to controls. Surprisingly, whereas in our conditions Dp1Yey mice travelled the same total distance in the OF as control mice, Dp1Yey/*Dyrk1a*$^{C/+}$ mice travelled significantly more distance (Fig 5A), suggesting that those mice are hyperactive. The percentage of time spent by the mice in the centre of the open field arena did not differ between genotypes (Fig 5B), suggesting normal anxiety-related behaviour. Mice heterozygous for *Dyrk1a* in glutamatergic neurons presented the same behaviour as control mice, indicating that removing only one copy of *Dyrk1a* is not enough to trigger decreased anxiety, as observed in the complete knockout of *Dyrk1a* in glutamatergic neurons (Figs 5B vs 4A). Analysis of working memory was done using the Y maze test. All the four groups of mice visited the same number of arms during the test, suggesting a normal locomotor activity (S5A Fig). On the other hand, Dp1Yey mice showed lower percentage of spontaneous alternation as compared to control mice (Fig 5C), confirming the phenotype already observed in previous studies [50,51]. This decreased performance was not restored by *Dyrk1a* normalization in glutamatergic neurons (Fig 5C). Haploinsufficiency of

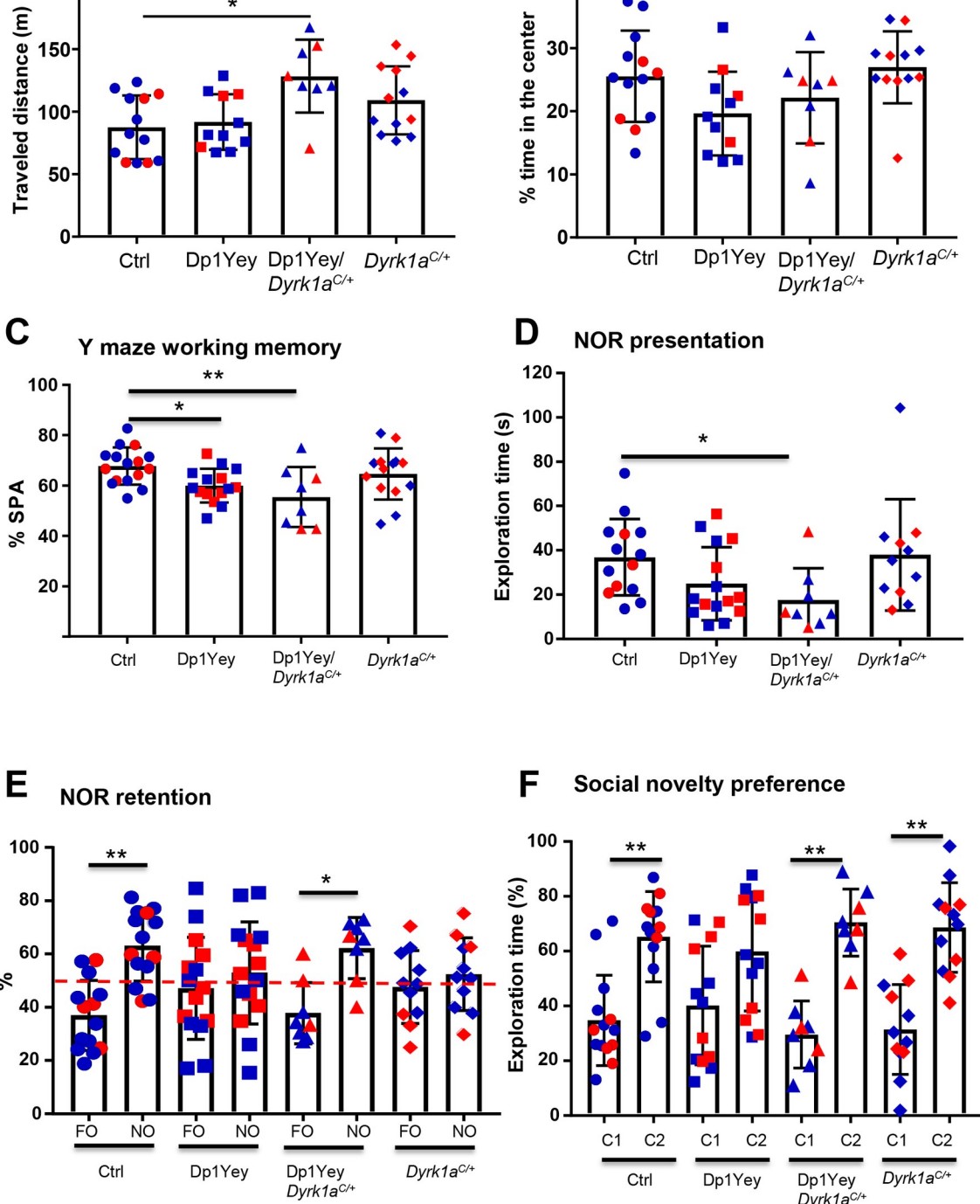

**Fig 5. Consequence of the normalization of *Dyrk1a* in glutamatergic neurons of Dp1Yey mice on animal cognition.** (A) The total distance travelled in the open field during a 30 min session is significantly increased in the Dp1Yey/*Dyrk1a*$^{C/+}$ mice compared to the control mice (Kruskal-Wallis One Way Analysis of Variance on Ranks, *p* = 0.009 with Dunn's post hoc multiple comparison procedures versus control, Dp1Yey/*Dyrk1*$^{C//+}$ vs control, *p = 0.007*). (B) Percentage time spent in the center of the OF does not vary between genotypes (One way ANOVA, F(3, 40) = 2.76, p = 0.054). (C) Percentage of spontaneous alternation of the mice during a 5 min session in a Y-maze. Lower percentage of alternation was found in Dp1Yey mice and in Dp1Yey/*Dyrk1a*$^{C/+}$ indicating a deficit in working memory in Dp1Yey mice that is not rescued in Dp1Yey/*Dyrk1a*$^{C/+}$ mice (One way ANOVA, Holm-Sidak method for multiple comparisons versus control group, F(3,49) = 4.3, *p* = 0.009, Dp1Yey vs control q = 2.48, *p* = 0.03, Dp1Yey/*Dyrk1a*$^{C/+}$ q = 3.24, *p* = 0.006). (D-E) Novel object recognition was assessed with 24 hours time laps. (D) Time spent exploring the two identical objects

during the first object presentation session was decreased in Dp1Yey/*Dyrk1a*$^{C/+}$ mice (Kruskal-Wallis One Way Analysis of Variance on Ranks, *p* = 0.02 with Dunn's post hoc multiple comparison procedures versus control, Dp1Yey/*Dyrk1a*$^{C//+}$ vs control, *p* = 0.02). (E) When introducing the novel object during the retention period, both control and Dp1Yey/*Dyrk1a*$^{C/+}$ lines spent significantly more time exploring the novel object than the familiar one (Paired t-test novel object vs familiar object, ctrl *p* = 0.003, Dp1Yey/*Dyrk1a*$^{C/+}$ *p* = 0.02), whereas *Dp1Yey* and *Dyrk1a*$^{C/+}$ mice did not (Paired t-test novel object vs familiar object, Dp1Yey *p* = 0.57, *Dyrk1a*$^{C/+}$ *p* = 0.56), revealing a significant deficit in memory for both Dp1Yey and *Dyrk1a*$^{C/+}$ mice which is rescued in Dp1Yey/*Dyrk1a*$^{C/+}$ mice. (F) Mice were tested for social novelty preference. All the genotypes but Dp1Yey spent significantly more time sniffing the new congener (Paired t-test congener vs empty cage, ctrl *p* = 0.004, Dp1Yey *p* = 0.13, Dp1Yey/*Dyrk1a*$^{C/+}$ *p* = 0.002, *Dyrk1a*$^{C/+}$ *p* = 0.002). Data are presented as point plots with mean ± SD. (n = 8–15 per genotype, *p<0.05, **p<0.01). Males (in blue) and females (in red) are pooled in the same graph as the statistical analyses did not reveal significant effect of sex.

*Dyrk1a* in those neurons, like the inactivation of the two copies of *Dyrk1a*, did not trigger any change in working memory (Fig 5C). We therefore also tested *Dyrk1a*$^{+/-}$ mice in the same test. *Dyrk1a*$^{+/-}$ animals showed the same activity (number of visited arms; S5B Fig) and the same level of alternation as their wild-type littermates, indicating a normal working memory (S5C Fig).

We further tested the mice in the NOR test for long term reference memory. Both control and Dp1Yey/*Dyrk1a*$^{C/+}$ mice showed a significant preferential exploration of the novel object during the retention trial (Fig 5E) whereas Dp1Yey and *Dyrk1a*$^{C/+}$ mice spent the same time on the two objects (Fig 5E). The deficit of novel object exploration during the retention phase in Dp1Yey and *Dyrk1a*$^{C/+}$ mice was not due to a lack of familiar object exploration during the presentation phase as both genotypes showed similar exploration times than control mice (Fig 5D). Only Dp1Yey/*Dyrk1a*$^{C/+}$ showed a slight decrease in object exploration compared to control mice during the presentation phase mice (Fig 5D), but this did not impair their retention capacity during the test phase. Hence, the deficit in object recognition in Dp1Yey mice could be rescued by normalization of *Dyrk1a* copy number in glutamatergic neurons and is also generated by the absence of one copy of the gene in the same neuronal cell line. In the fear conditioning test, all genotypes showed more freezing during the context phase after conditioning than during the habituation phase and no difference was observed between genotypes in the context response (S5D Fig). In the sociability 3-chambers test, all the four groups of mice showed preference for the cage containing the mouse rather than the empty cage (S5E Fig). No difference was found between the four groups in the total amount of time spent sniffing the cage containing the congener (S5F Fig). Hence, by contrast to *Dyrk1a*$^{C/C}$ mice, *Dyrk1a*$^{C/+}$ mice do not present decreased social exploratory behaviour. Dp1Yey mice did not spend significantly more time with the novel mouse compared to the familiar one, indicating no preference for social novelty (Fig 5F). This phenotype was rescued by returning to two copies of *Dyrk1a* in glutamatergic neurons (Fig 5F). *Dyrk1a*$^{C/+}$ mice also showed preference for social novelty (Fig 5F).

Hence, both increase in *Dyrk1a* copy number in glutamatergic neurons of trisomic mice and haploinsufficiency of *Dyrk1a* in glutamatergic neurons impact explicit memory supporting a key role of *Dyrk1a* in glutamatergic function as a modulator of explicit memory, but other functions require normalization of *Dyrk1a* in other cell types to be restored.

## Proteomic analysis suggests an impact of *Dyrk1a* gene dosage on synaptic activity

To examine the contribution of DYRK1A in molecular pathways linked to the cognitive phenotypes associated to T21 in the glutamatergic neurons, we performed proteomic profiling of the hippocampus of control, Dp1Yey, Dp1Yey/*Dyrk1a*$^{C/+}$ and *Dyrk1a*$^{C/+}$ mice. We identified 63 proteins that were up-regulated and 16 that were down-regulated in the hippocampi of Dp1Yey mice compared with controls. Among those, 40 of the up-regulated and 12 of the

down-regulated proteins were back to control levels in Dp1Yey/*Dyrk1a*$^{C/+}$ hippocampi, while one up-regulated protein in Dp1Yey was down-regulated in Dp1Yey/*Dyrk1a*$^{C/+}$ and 4 down-regulated proteins in Dp1Yey were up-regulated in Dp1Yey/*Dyrk1a*$^{C/+}$ mice. We found 51 up-regulated and 7 down-regulated proteins in the hippocampus of *Dyrk1a*$^{C/+}$ mice. Eleven of those proteins (CAMK2A, ATP6V1C1, DPP3, ERGIC1, GPM6A, CENPV, RPS28, AGAP2, SNX6, ABCA1, BRK1) were also deregulated in Dp1Yey and back to normal level in Dp1Yey/*Dyrk1a*$^{C/+}$, suggesting that they are impacted by *Dyrk1a* copy number in glutamatergic neurons (Fig 6A and S5 Table). We performed GO enrichment analysis on the list of deregulated proteins using the ToppCluster website, selecting a Bonferroni cut-off of P<0.05. Enrichment analysis indicates that pathways and GO components that are mostly affected by *Dyrk1a* gene dosage are synaptic, dendritic and axonal components (Fig 6B and 6C; S6 Table). Normalization of *Dyrk1a* copy number in the glutamatergic neurons did not rescue specific pathways but had a more global effect with 50 to 80% of the proteins present in each Dp1Yey enriched GO returning to normal amount in Dp1Yey/*Dyrk1a*$^{C/+}$ mice (Fig 6B). Interestingly, decreased *Dyrk1a* gene dosage was found to impact pre-synaptic proteins as observed in the transcriptome of *Dyrk1a*$^{C/C}$ hippocampi, whereas increased *Dyrk1a* gene dosage was associated with the post-synapse and growth cone (Fig 6C). Proteins enriched in the hippocampus of *Dyrk1a*$^{C/+}$ mice were linked to translational activity whereas increased *Dyrk1a* gene dosage was associated to ATPase activity (Fig 6C).

### Interaction of DYRK1A with post-synaptic proteins

Behavioural and proteomic analyses suggest a direct impact of DYRK1A at the glutamatergic synapse. Previous work from our laboratory already demonstrated a role of DYRK1A at the presynapse by showing interaction of DYRK1A with SYN1, a neuronal phosphoprotein associating with the cytoplasmic surface of the presynaptic vesicles and tethering them to the actin cytoskeleton [52,53], and with CAMK2 that was previously shown to phosphorylate SYN1 leading to the release of the vesicle pool [54–56]. Moreover, we also found that SYN1 was phosphorylated by DYRK1A on its S551 residue *in vitro* and *in vivo*, highlighting the role of DYRK1A in SYN1-dependent presynaptic vesicle trafficking [56]. CAMK2, deregulated in our proteomic analysis, is also present in the glutamatergic postsynapse and has a major role in the molecular cascade leading to LTP [57]. To investigate a potential role of DYRK1A at the post-synapse, we looked at DYRK1A protein interaction with CAMK2 and key proteins of the post-synaptic density complex (PSD), GLUN2B (NR2B), PSD95 and SYNGAP. We carried co-immunoprecipitation (co-IP) experiments with adult mouse brain lysates using antibodies against DYRK1A and these proteins and using GAPDH as a negative control. We found CAMK2A, NR2B and PSD95 present in the immunoprecipitates (IPs) of DYRK1A, while DYRK1A was found in the IPs of NR2B, PSD95 and SYNGAP (Fig 6D), showing that these proteins interact together.

### Discussion

Complete *Dyrk1a* inactivation leads to early embryonic lethality with homozygous null *Dyrk1a* mice presenting drastic developmental growth delay with smaller brain vesicles, hindering the investigation of *Dyrk1a* function in the brain [32]. We therefore used a conditional knockout strategy to analyse *Dyrk1a* function in glutamatergic neurons. Contrary to *Dyrk1a* inactivation in cortical progenitor cells and glutamatergic postmitotic neurons leading to a lack of corpus callosum and lateral cortex with the death of the animal after birth [58], inactivation of *Dyrk1a* in postnatal glutamatergic neurons results in a milder phenotype. *Dyrk1a*$^{C/C}$ mice are viable and present a mild microcephaly with a 10% brain weight reduction and a

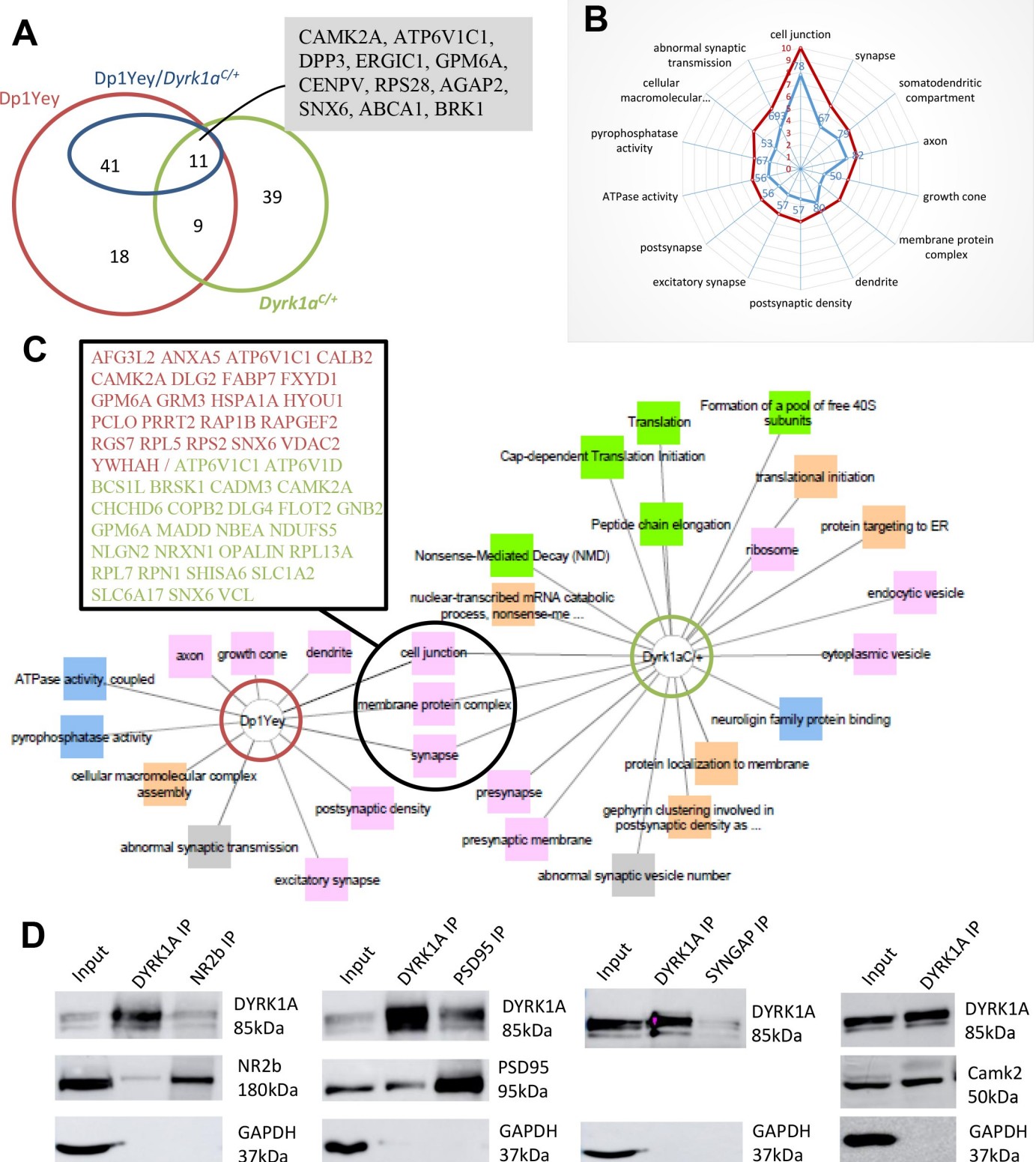

**Fig 6. Proteomic analysis (A)** Venn diagram showing the numbers of deregulated proteins in the different mouse models. The numbers of proteins shown for the Dp1Yey/*Dyrk1a*^C/+ model (in dark blue) correspond to proteins that were deregulated in the Dp1Yey and back to normal levels in this model (proteins that are regulated by Dyrk1a in the trisomy). Proteins deregulated by both *Dyrk1a* up and down-regulation (common to Dp1Yey, Dp1Yey/*Dyrk1a*^C/+ and *Dyrk1a*^C/+) are listed in the grey shaded box. **(B)** Radar plots of GO terms that are mostly enriched in the Dp1Yey model (in red with scale bar corresponding to the log of the p-

value) and of the proportion of the proteins found deregulated in Dp1Yey which amount is normalized by the return in 2 copies in the Dp1Yey/*Dyrk1a*$^{C/+}$ model (in blue with scale bar corresponding to the % of Dp1Yey deregulated protein back to normal levels). (C) Visual representation of the GO enrichments for the deregulated proteins in Dp1Yey and *Dyrk1a*$^{C/+}$ hippocampi with connection between common terms. The different categories of GO are represented by different colors: pink for Cellular components", blue for "Molecular functions", green for "Pathways", orange for "Biological processes" and grey for "Phenotypes". The list of proteins in the box corresponds to proteins present in the common deregulated GO terms (in red deregulated proteins in Dp1Yey and in green proteins deregulated in *Dyrk1a*$^{C/+}$). (D) Western blots of DYRK1A, NR2B, PSD95, SYNGAP and CAMK2 proteins following IPs of wild-type mice brain extracts. We found in NR2B, PSD95 and CAMK2 in the IPs of DYRK1A. We also detected DYRK1A in the IPs of NR2B, PSD95 and SYNGAP.

principal impact on cortical thickness in the pure B6N genetic background. In comparison, *Dyrk1a*$^{+/-}$ mice from a mixed B6/129OLA genetic background showed 30% brain reduction and the cortical size reduction was also associated with increased cell density [31,39], suggesting a reduction of neuronal processes as observed in the neocortex of *Dyrk1a*$^{+/-}$ mice and in primary neurons in culture [31,59]. Surprisingly, mice with heterozygous inactivation of *Dyrk1a* in developing excitatory neurons of the cortex (*Dyrk1a*$^{Emx1/+}$) presented decreased cortical thickness and increased cell density due to smaller excitatory neurons only at birth, while adult microcephaly was associated to fewer neurons due to increased apoptosis leading to postnatal cell death [58]. This suggests that microcephaly observed in MRD7 results from different impacts of DYRK1A on brain neurogenesis during embryonic and postnatal development, with *Dyrk1a*$^{C/C}$ brain revealing the impact of DYRK1A on postnatal neuronal morphogenesis.

Beside its role in the perception and processing of sensory information, the somatosensory cortex is also implicated in the integration of high order information and participates in linking perception of socially relevant stimuli to motivation, emotion and perception [60], features that are strongly perturbated in ASD. We found that *Dyrk1a*$^{C/C}$ mice spent less time in social contact than control mice, a phenotype also observed in *Dyrk1a*$^{+/-}$ and *Dyrk1a*$^{Emx1/+}$ animals [58,61,62], suggesting a role for DYRK1A in cortical glutamatergic neurons in ASD-like deficits observed in persons with *DYRK1A* mutations. However, postnatal inactivation of *Dyrk1a* in glutamatergic neurons was not sufficient to trigger susceptibility to epileptic activity as observed in *Dyrk1a*$^{+/-}$ mice [61,63], suggesting that absence of DYRK1A in glutamatergic neurons is not responsible for this phenotype.

DYRK1A deficit in glutamatergic neurons had an impact on mouse emotional behaviour. *Dyrk1a*$^{C/C}$ mice were less anxious, spending more time in the centre of the OF and in the open arms of the EPM, and showed decreased freezing performance in the fear-related contextual. Encoding of contextual fear conditioning memory requires the hippocampal-amygdala circuit together with frontal cortices region [64]. Deficit in contextual fear behaviour is often attributed to a defective hippocampal-to-basolateral amygdala transmission as a result of either a deficit in glutamatergic projections or deficit in excitatory activity [65], although it was shown that context freezing one day after conditioning partly occurs through reactivation of neuronal ensembles in the primary sensory cortex [66]. Change in emotional behaviour in *Dyrk1a*$^{C/C}$ mice is not the result of an intrinsic hyperactivity, as the mice did not present increased locomotor activity either spontaneous (circadian activity) or novelty induced (OF). Hence, contrary to the hypoactivity induced by full *Dyrk1a* haploinsufficiency [67], absence of DYRK1A in glutamatergic neurons does not impact mouse locomotor activity. Thermal pain sensitivity was altered in *Dyrk1a*$^{C/C}$ mice that were more sensitive to heat. This higher nociception response was also observed in *Dyrk1a*$^{+/-}$ mice, suggesting that DYRK1A has an impact on central processes involved in the control of pain sensitivity. The glutamatergic system takes part in the nociceptive circuits and activation of the expression of IEGs, whose expression was found increased in *Dyrk1a*$^{C/C}$ mice, has been shown to be part of long-term events triggered in neuroadaptation to pain in those circuits [68]. Interestingly, we and our collaborators observed a

decreased in the expression of some of these IEGs (*Npas4*, *Arc*, *c-Fos* and *Fosb*) in the hippocampi of Tg(Dyrk1a) and of the trisomic mouse models Dp1Rhr and Ts65Dn [50,69]. IEGs are also believed to be crucial in the formation of long-term memory which we also found impacted in *Dyrk1a*$^{C/C}$ and trisomic mice [70]. Moreover, our meta-analysis of the transcriptomic data of hippocampi from five DS mouse models carrying Mmu16 segmental duplications and a transgenic model overexpressing *Dyrk1a* revealed regulatory protein networks centred around six protein hubs, among which were DYRK1A itself and NPAS4 [50]. *Npas4* is a neuron-specific gene and is present in both excitatory and inhibitory neurons, activating distinct programs of late-response genes promoting inhibition onto excitatory neurons and excitation on inhibitory neurons [49]. But we did not found major changes in the expression of late response genes targeted by NPAS4 (S3 Table) [49]. This could be due to experimental bias as the transcriptomic analysis was done using RNA extracts from whole hippocampi containing different cell populations. This heterogeneity can hinder glutamatergic-specific expression of the late-response genes. Even though IEGs are well known markers to measure neuronal activity during cognitive stimulation, their impact in cognitive processes affected in cognitive deficit disorders are unknown and we have no explanation for IEGs overexpression in the hippocampus of *Dyrk1a*$^{C/C}$ animals. In our analysis, IEG expression changes have been observed in "naïve" mice that were not subjected to any exercise or behavioural test. It would be therefore also interesting to analyse the expression of IEGs and late-response genes in the mice after induction of neuronal activity, as it was explored for activation of Arc mRNA transcription in pyramidal neurons of the CA1 region of the hippocampus in Ts65Dn mice [71].

Cognitive deficit is often associated with impairment of memory capacity. Individuals with DS are especially affected in their explicit long-term memory abilities with a particular impairment in the visuo-perceptual processing [72]. Ts65Dn and Dp1Yey DS mouse models as well as Tg(Dyrk1a) and *Dyrk1a*$^{+/-}$ have been shown to have impaired long-term object recognition memory [24,50,56,73]. Novel object recognition requires a network of medial-temporal lobe regions including the hippocampus and the perirhinal-parahippocampal-entothinal and insular cortices [74]. Deficit in NOR memory has been linked to perturbation between excitatory and inhibitory neurotransmission and excessive GABAergic inhibition has been proposed as the major cause, with glutamatergic deficit being the consequence of over-inhibition of the NMDA receptors resulting in deficit of LTP and memory [75]. We show here that correcting *Dyrk1a* gene copy number in glutamatergic neurons is sufficient to rescue explicit long-term memory in Dp1Yey/*Dyrk1a*$^{C/+}$ mice. Moreover, DYRK1A shortage in glutamatergic neurons is also sufficient to trigger long-term memory deficit in both *Dyrk1a*$^{C/+}$ and *Dyrk1a*$^{C/C}$ mice. Our finding outlines the glutamatergic deficit as a distinct alteration with *Dyrk1a* overexpression playing a key role in glutamatergic dysfunction and GABA-mediated over-inhibition combining with it to produce the full DS cognitive deficit. This also raises the question of the role of *Dyrk1a* overexpression in GABAergic neurons as other trisomic genes are also potential candidates for neuronal dysfunction. For example, overexpression of *Girk2* leads to increase in GABA$_A$-mediated GIRK currents in hippocampal neuronal cultures, affecting the balance between excitatory and inhibitory transmission [76,77]. TgDyrk1a mice have been shown to have bidirectional changes in synaptic strength with elevated LTP, reduced LTD [22] and dysregulated NMDA-receptor mediated calcium signalling [78]. Furthermore, normalization of *Dyrk1a* expression in the hippocampus of Ts65Dn mice can partially restore the deficit of LTP in the CA1 of Ts65Dn mice. As Dp1Yey mice show similar hippocampal LTP deficit [79], it would be interesting to see if *Dyrk1a* normalisation in the glutamatergic neurons could restore LTP in Dp1Yey/*Dyrk1a*$^{C/+}$ mice.

The finding of a cell-autonomous impact of DYRK1A in glutamatergic neurons on long-term memory function is supported by the impact of increased *Dyrk1a* gene dosage in

glutamatergic neurons on the amount of glutamatergic post-synaptic proteins. Hence, among enriched proteins in the hippocampus of Dp1Yey mice that turned back to normal in the hippocampus of Dp1Yey/*Dyrk1a*$^{C/+}$ mice, we found CAMK2A, a subunit of the calcium/calmodulin-dependent protein kinase II (CAMK2) which plays a critical role in LTP by regulating ionotropic glutamate receptors at postsynaptic densities, GPM6A, a neuronal membrane glycoprotein involved in neuronal plasticity, regulation of endocytosis and intracellular trafficking of G-protein-coupled receptors [80], the GRM3 G-protein-coupled metabotropic glutamate receptor, DLG2, a member of the postsynaptic protein scaffold of excitatory synapses interacting with the cytoplasmic tail of NMDA receptors [81] and the intracellular calcium-binding protein CALB2 functioning as a modulator of neuronal excitability [82]. Previous work done in our laboratory found several proteins from the PSD that were hyperphosphorylated in mice with three copies of *Dyrk1a* (TgDyrk1a) [39] and dephosphorylated by TgDyrk1a mouse treatment with a DYRK1A inhibitor [56]. Among those proteins, the NR2B is a subunit of the glutamatergic postsynaptic NMDA receptor which play a pivotal role in excitatory synaptic transmission. This result was validated by our Co-IP experiments showing an interaction between DYRK1A and NR2B. NR2B subunits are expressed in the neocortex and hippocampus [83–85]. NMDA receptors in the mature hippocampus consist of two NR1 subunits associated with either two NR2A, two NR2B or one of each subunits [86,87] and different forms of synaptic plasticity have been associated to different types of NMDA receptors [88–90]. Hence, in addition to its interaction with the NR1/NR2A-type of receptors [91], we also point out an association with NR1/NR2B receptors. Moreover, interaction between DYRK1A and the PSD proteins PSD95, CAMK2 and SYNGAP, detected by coIP, strongly suggests a role of DYRK1A at the glutamatergic postsynapse. In addition, absence of the GLUR1 subunit of the AMPA receptor in the IP of DYRK1A indicates that DYRK1A interact most specifically with the NMDA-PSD complex. Altogether, this strongly suggests an implication of DYRK1A at the glutamatergic post-synapse, somehow supporting its involvement in long-term memory formation.

*Dyrk1a*$^{C/C}$ transcriptomic analysis also revealed a down-regulation of genes for proteins involved in presynaptic vesicle cycle, implicating DYRK1A in neurotransmitter release. Among the deregulated genes, we found *Amphiphysin* (*Amph*), which its protein is a known target of DYRK1A [92], and *Synapsin 2* (*Syn2*), which paralog SYNAPSIN 1 was found to be hyperphosphorylated by DYRK1A overexpression in TgDyrk1a mice [56]. Moreover, DYRK1A was found to phosphorylate MUNC18-1 [93], which interacts with the SNARE complex protein Syntaxin 1A, whose transcripts are also decreased in the hippocampus of *Dyrk1a*$^{C/C}$ mice and which was found to be one of the six hubs connecting the major subnetwork biological cascades found deregulated in DS models [50]. Our results together with others [92,94,95] point at a role of DYRK1A in the glutamatergic presynapse in the control of neurotransmitter release through synaptic vesicles exocytosis and vesicles recycling processes.

Taking advantage of a conditional allele for *Dyrk1a* inactivation, we were able to associate *Dyrk1a* gene dosage changes in glutamatergic neurons to specific cognitive phenotypes and molecular modifications and demonstrated a major impact of *Dyrk1a* dose change at the glutamatergic synapse on long-term explicit memory while no impact was observed for motor activity, short-term working memory and susceptibility to epilepsy. Further analysis of DYRK1A impact on other neurons, such as GABAergic ones, will be necessary to understand how DYRK1A perturbs the excitatory/inhibitory pathways, resulting in the full DS and MRD7 cognitive deficits.

## Materials and methods

### Ethic statement

Animal research was performed in agreement with the EC directive 2010/63/UE86/609/CEE and in compliance with the animal welfare policies of the French Ministry of Agriculture (law 87 848). Behavioural experiments were approved by the local animal care, use and ethic committee of the IGBMC (Com'Eth, no.17, APAFIS 2012–069). The PTZ-induced seizures protocol received the accreditation number APAFIS#6321.

### Mouse lines

The Dp(16Lipi-Zbtb21)1Yey (Dp1Yey) line was created by Yu and collaborators [96] and bears a 22.6 Mb segmental duplication of the *Lipi-Zfp295* fragment of murine chromosome 16 syntenic to Hsa21 [79,96]. The transgenic Tg(Camk2-Cre)4Gsc line [33,97] expressing the Cre recombinase under the control of the *Camk2a* promoter was used to inactivate the targeted conditional knockout allele in glutamatergic neurons of the cortex and hippocampus after birth. The *Dyrk1*$^{tm1.ICS}$ conditional knockout (noted *Dyrk1a*$^{cKO}$) was generated at the PHENOMIN- ICS (Institut Clinique de la Souris; Illkirch, France; www.phenomin.fr) in the frame of the Gencodys consortium (http://www.gencodys.eu/). The targeting vector was constructed as follows. A 1096 bps fragment encompassing exon 7 (ENSMUSE00001246185) was amplified by PCR (from BAC RP23-115D20 genomic DNA) and subcloned in an MCI proprietary vector. This MCI vector contains a LoxP site as well as a floxed and flipped Neomycin resistance cassette. A 3.8 kb fragment corresponding to the 3' homology arm and 4.1 kb fragment corresponding to the 5' homology arms were amplified by PCR and subcloned in step1 plasmid to generate the final targeting construct. The linearized construct was electroporated in C57BL/6N (B6N) mouse embryonic stem (ES) cells. After selection, targeted clones were identified by PCR using external primers and further confirmed by Southern blot with a Neo probe (5' and 3' digests) as well as a 5' external probe. Two positive ES clones were injected into BALB/cN blastocysts. Resulting male chimeras were bred with Flp deleter females previously backcrossed in a C57BL/6N [98] (PMID: 10835623). Germline transmission of the conditional allele was obtained (Fig 1B). The Flp transgene was segregated by a further breeding step. Dyrk1a constitutive heterozygous knockout (*Dyrk1a*$^{+/-}$) mice were generated by mating *Dyrk1a*$^{cKO/+}$ animals with the *Hprt1*$^{tm1(Cre)Mnn}$ line (http://jaxmice.jax.org/strain/004302.html) to obtain a germline deletion of *Dyrk1a*. As *Dyrk1a*$^{+/-}$ animals could not be generated on a pure background due to lethality, *Hprt1*$^{tm1(Cre)Mnn}$ females and *Dyrk1a*$^{cKO/+}$ males were first crossed with the C$_3$H/HeH line (C3B line; congenic line for the BALB/c allele at the *Pde6b* locus [99] to generate a F1 B6C3B mixed background before being mated together. *Dyrk1a*$^{+/-}$ were kept on this mixed background by crossing *Dyrk1a*$^{+/-}$ with wild-type F1 B6C3B animals. Validation of the *Dyrk1a*$^{+/-}$ model is presented in S6 Fig.

Animals for behavioural analysis were obtained by first mating Tg(Camk2-Cre)4Gsc males with Dyrk1a$^{cKO/cKO}$ females to generate *Dyrk1a*$^{C/+}$ animals. Those animals were tested for the absence of the Dyrk1a KO allele by genotyping in order to avoid to use animals that could have *Dyrk1a* germline recombination. *Dyrk1a*$^{C/+}$ males were further mated with *Dyrk1a*$^{cKO/cKO}$ females to obtain *Dyrk1a*$^{C/C}$ animals as well as *Dyrk1a*$^{cKO/+}$ and *Dyrk1a*$^{cKO/cKO}$ that were used as controls (S10 Table). *Dyrk1a*$^{C/+}$ females obtained during the first round of mating were further mated with Dp1Yey males in order to produce mice: Dp1Yey (trisomic, for the *Lipi-Zfp295* fragment containing the *Dyrk1a* gene), Dp1Yey/*Dyrk1a*$^{C/+}$ (trisomic for the *Lipi-Zfp295* fragment but containing only two copies of *Dyrk1a* in glutamatergic neurons), *Dyrk1a*$^{C/+}$ (containing only one copy of *Dyrk1a* in the glutamatergic neurons) and *Dyrk1a*$^{C/C}$ (knocked out for *Dyrk1a* in glutamatergic neurons). Wild-type, *Dyrk1a*$^{cKO/+}$ and *Dyrk1a*$^{cKO/cKO}$ mice were used as disomic controls (S7 Table).

For the genotyping of the mice and identification of the *Dyrk1a* knockout allele in the brain, genomic DNA was isolated from tail and different organ biopsies using the NaCl precipitation technique. 50–100 ng of genomic DNA was used for PCR. Primers used for the identification of each allele and size of PCR products are described in Fig 1 and S8 Table. Details on the genotyping protocol used here are published [100].

The mice were housed in groups (2–4 per cage) and were maintained under specific pathogen-free (SPF) conditions and were treated in compliance with the animal welfare policies of the French Ministry of Agriculture 133 (law 87 848).

## Mouse RT droplet digital PCR (ddPCR)

Total RNA was extracted from frozen brain tissues of young adult mice (males and females, 3 months old) (cerebellum, cortex, striatum, hippocampus and thalamus/hypothalamus) of five wt and four *Dyrk1a*$^{C/C}$ mice as described in Lindner *et al*. 2020 [101]. For ddPCR, all primers were designed and synthesized as described in Lindner *et al*. 2020 [101] excepting Universal Probe Library probe used to *Dyrk1a* mRNA which is provided by Roche. *Dyrk1a* and *Hprt* primers and probes sequences are given in S8 Table. RNA reverse transcription, droplet generation, PCR amplification, droplets quantification and analysis are also described in Lindner *et al* 2020 [101]. We presented the results as a ratio of the mean of *Dyrk1a* RNA transcript in *Dyrk1a*$^{C/C}$ tissue normalized to the mean of *Dyrk1a* RNA transcript in wt tissue. Experiments were performed following dMIQE guidelines for reporting ddPCR experiments (S9 Table) [101,102].

## Western blot analysis

Twenty-five micrograms of total protein extracts from hippocampi (young adult males and females 3 months old, n = 3 per genotype) were electrophoretically separated in SDS-polyacrylamide gels (10%) and transferred to a nitrocellulose membrane (100V, 2h at room temperature). Non-specific binding sites were blocked with 5% skimmed milk in Tween20 0.1% Tris buffer saline 1h at room temperature. Immunostaining was carried out with a mouse monoclonal anti-Dyrk1a (Abnova, H00001859-M01) and an anti-Gapdh antibodies (Thermo-Fisher, MA5-15738), followed by secondary anti-mouse IgG conjugated with horseradish peroxidase (DAKO). The immunoreactions were visualized by ECL chemiluminescence system (Amersham) with the Amersham Imager 600. Semi-quantitative analysis was performed using ImageJ software (W. Rasband, NIH; http://rsb.info.nih.gov/ij/).

## Immunohistological analysis

Young adult female mice (3 months old) for immunohistochemistry and young adult males (2-month-old) for immunofluorescence were deeply anesthetized with sodium pentobarbital and perfused intracardially with 30 ml PBS followed by 30 ml 4% paraformaldehyde in PBS. Brains were removed from the skull and immersed in the same fixative overnight. After rinsing with PBS, the brains were transferred into 70% ethanol until paraffin inclusion. For inclusion, brains were dehydrated and embedded in paraffin. Serial 10 μm sections were made with a microtome.

Brain sections were stained using the myelin-specific dye luxol fast blue and the Nissl staining cresyl violet. Briefly, brain sections were deparaffinised, rehydrated and incubated in 0.1% luxol fast blue (95% alcohol and 0.5% acetic acid) solution at room temperature overnight. After rinsing excess stain with 95% ethanol and deionized water, the slides were placed in 0.05% lithium carbonate solution for 10 seconds followed by 70% ethanol for 5 seconds. They were then rinsed in deionized water until the colourless grey matter contrasted with the blue-

green white matter. Sections were then stained in 0.1% cresyl violet acetate solution for 5 minutes at 56˚C in a water bath, rinsed in deionized water and quickly in 100% ethanol. Sections were dried, cleared and mounted.

Immunohistology was performed using a standard protocol. After deparaffinization, rehydration and antigen retrieval (10 mM citric acid, 0.05% Tween 20, pH6.0) for 45 min in a 94˚C water bath, sections were incubated in a blocking solution (0.05% Tween20, 5% Horse serum) for 1 hour at room temperature. Sections were then incubated at 4˚C overnight with the primary antibodies (Mouse anti-DYRK1A: Abnova, Cat. N. H00001859-M01; Rabbit anti-CAMK2A: Molecular probes, PA5-14315). After washing, sections were incubated with anti-Mouse and anti-Rabbit Alexa fluorescent 546 or 488 secondary antibodies for detection (Abcam, Paris). Control immunostainings were also performed with secondary antibodies only. The sections were mounted with Mowiol mounting medium (0.1M Tris (pH8.5, 25% glycerol, 10% w/v Mowiol 4–88 (Citifluor)) containing DAPI (5 μg/ml) and images were acquired using Hamamatsu Nanozoomer 2.0 (Hamamatsu, Hamamatsu City, Japan) and a Leica Upright fluorescent microscope (Leica Microsystems, Heidelberg).

Immunohistochemistry was performed using a standard protocol. Briefly, antigen retrieval was performed by heating the slides in Tris/EDTA buffer (10 mM Tris Base, 1 mM EDTA, 0.05% Tween 20, pH 9.0) or citrate buffer 1 mM, pH6 for 45 min in a 94˚C water bath. Then, the sections were quenched in 0.3% oxygen peroxide solution for 20 min and blocked with 10% normal horse serum and 0.1% Triton X-100 in 1× PBS for 1 h at room temperature. The sections were incubated overnight at 4˚C with primary antibodies (rabbit anti-Olig2, 1:500, Santa Cruz sc-48817; Rabbit anti NeuN, 1:500 Sigma, ZRB377; Rabbit anti-S100b, 1:1000, Sigma HPA015768). which was detected by incubating the sections with secondary biotinylated antibodies (Life Technologies, France) for 2 h at room temperature and then with an avidin-biotin complex at 37˚C for 30 min. Dark coloration was developed with diaminobenzidine tetrahydrochloride and the sections were mounted with aqueous mounting medium (Agilent, Les Ulis, France).

## Morphometric analysis and cell counting

Morphometric analysis was performed on three *Dyrk1a*$^{C/C}$ and three control adult females mice (3 months old) based on the standard operating procedures for morphological phenotyping of the mouse brain using basic histology [103]. Coronal sections for morphometric analysis and cell counting were carefully selected using specific anatomical landmarks that enables to precisely select slides that are at the same antero-posterior level (sections showing upper and lower arms of the dentate gyrus of the same length). Surface and cortical thickness measurements as well as cell counting were conducted on scanned images using Hamamatsu Nanozoomer 2.0 from luxol fast blue/cresyl violet-stained and NeuN-labeled and S100b-labeled sections around Bregma -1.5 mm (Paxinos adult mouse brain atlas, Franklin and Paxinos, 1997). TIFF files were opened in ImageJ with the following settings: 9 decimal places (using the panel Analyze/Set Measurements) and "cm" as unit length (using Analyze/Set Scale). The polygone selection tool was used to measure area and the straight-line tool was selected to measure length. The thickness of the different cortical layers (layer I to layer VI) were estimated in the somatosensory cortex based on the shape and density of the neurons on these different layers. Cell count performed in the somatosensory cortex was done within a counting frame of 0.1 cm. Cell count performed in the CA1 was done within a counting frame of 0.04 cm width). Olig2+-positive cells within the corpus callosum were counted by measuring a distance of 1 mm from the midline of the brain and selecting the corpus callosum area underneath. Cell count was done manually. It should be noted that cell counts were not acquired

using randomly sampled unbiased stereological standards and therefore cannot necessarily be extrapolated to the entire cortex and hippocampus.

## RNA-seq libraries and analysis

Total RNA was Trizol-extracted from 2 wild-type and 2 Dyrk1a$^{C/C}$ frozen P30 young adult male hippocampi, stage at which the brain is fully developed. RNA was treated with DNase (Qiagen) and purified on the RNeasy MinElute Cleanup Kit (Qiagen). 2 µg of total RNA were treated with the Ribo-Zero rRNA Removal Kit (Human/Mouse/Rat; Illumina). Depleted RNA was precipitated 1h at -80°C in three volumes of ethanol plus 1 µg of glycogen. RNA was then washed and resuspended in 36 µl of RNAse free water. RNA fragmentation buffer (NEBNext, New England Biolabs, Evry, France) was added to the solution and the RNA was fragmented by incubation at 95°C for 3 min. cDNA first strand synthesis was performed with random hexamer primers and cDNA second strand synthesis was performed with dUTPs, to ensure strand specificity. The RNA-seq library was synthetized with KAPA Hyper prep kit (Kapa Biosystems, Wilmington, MA, USA): a treatment with USER enzyme (NEB, M5505L) was added to digest the unspecific strand.

The libraries were pooled (4/lane) on an Illumina HiSeq. 2000. Libraries were sequenced (50 cycles, single-end) yielding on average 40 million mapped reads. RNA-Seq libraries were mapped with GSNAP (version 2015-06-23) against mm9 mouse RefSeq annotations updated to the 28/7/2015.

DESeq 2 (v1.14) was used to perform statistical comparisons. All the enrichment analysis were made from standard hypergeometric tests with benjamini or bonferroni correction. The markers of hippocampal cell types were obtained from [104] and the common background genes were evaluated prior to the enrichment (hypergeometric test). GO annotations were updated to 25/6/2015.

## Proteomic analysis

Fourty micrograms of total protein extracts from hippocampus of 4 months old males coming from cohorts that underwent behavioural analysis (4 controls, 5 Dp1Yey, 4 Dp1Yey; *Dyrk1a*$^{C/+}$ and 5 *Dyrk1a*$^{C/+}$) were used for the preparation. Samples were precipitated, reduced, alkylated and digested with LysC and trypsin at 37°C overnight. 10 µg of each sample were then labeled with TMT isobaric tags, pooled, desalted on a C18 spin-column and dried on a speed-vacuum before nanoLC-MS/MS analysis. Samples were separated on a C18 Accucore nano-column (75 µm ID x50 cm, 2.6 µm, 150 Å, Thermo Fisher Scientific) coupled in line with an Orbitrap ELITE mass spectrometer (Thermo Scientific, San Jose, California). Samples were analyzed in a Top15 HCD (High Collision Dissociation) mass spectrometry on 8h gradient. Data were processed by database searching using SequestHT (Thermo Fisher Scientific) with Proteome Discoverer 1.4 software (Thermo Fisher Scientific) against a mouse Swissprot database (release 2015–03). Peptides were filtered at 5% false discovery rate (FDR) and one peptide in rank 1. Protein quantitation (ratio of the intensity of the fragmented tag in sample "x" to the intensity of the fragmented tag in one control (disomic) sample used as the reference) was performed with reporter ions quantifier node in Proteome Discoverer 1.4 software with integration tolerance of 20 ppm, and the purity correction factor were applied according to the manufacturer's instructions. A scaling factor normalization method was used in order to make sample ratios comparable. Ratios were normalized by calculating the mean of all the peptide ratios in one sample, calculating a scaling factor (sf = mean [ratio control ref]/mean [ratio sample x]) for each sample and multiplying each ratio by the sf. Data were filtered with the following criteria: minimum number of peptide ratios used to calculate the protein ratio equal to 2; variability of

the peptide ratios <20%; ratio of Dp1Yey and *Dyrk1a*[C/+] samples compared to mean of disomic controls, x>1.2 or x<0.8 and ratio of Dp1Yey;*Dyrk1a*[C/+] samples compared to mean of disomic controls 0.8<x<1.2 among proteins that were selected as deregulated in Dp1Yey. GO enrichment was calculated in the ToppCluster website (https://toppcluster.cchmc.org/), looking at enrichment within the following features: Molecular functions, Biological processes, Cellular components, Phenotypes and Pathways, and using a Bonferroni correction cut-off of P<0.05. The results of the enrichments can be found in the S6 Table.

## Co-immunoprecipitation

Immunoprecipitations were performed on fresh half brains of 3-month-old wild-type male mice. Brains were dissected and lysed in 1.2 ml RIPA lysis buffer (Santa-Cruz Biotechnology, France) using Precellys homogenizer tubes (Bertin Instruments, Montigny-le-Bretonneux, France). After centrifugation at 2800 g for 2×15 s, 1 ml brain extract was incubated with 2 μg of antibody of interest at 4°C for 1 h under gentle rotation. An aliquot of the remaining supernatant was kept for further immunoblotting as homogenate control. Then, 20 μl protein G agarose beads, previously washed three times with bead buffer, were added to the mix and gently rotated at 4°C for 30 min. After a 1 min spin at 10,000 g and removal of the supernatant, the pelleted immune complexes were washed three times with bead buffer before WB analysis with appropriate antibodies directed against DYRK1A (H00001859 M01, Interchim; 1:1000), NMDAR2B (Abcam, #ab65783), PSD95 (ab18258, Abcam, France; 1:1000), CAMK2A (PA5-14315, Thermo Fisher Scientific; 1:1000), SYNGAP (sc-8572, Santa Cruz biotechnologies; 1:5000) and GAPDH (MA5-15738, Thermo Fisher Scientific; 1:3000). Immunoblots were revealed with Clarity Western ECL Substrate (Bio-Rad).

## Mouse behavioural analysis

A series of behavioural experiments were conducted in adult mice. Behavioural analyses of *Dyrk1a*[C/C] animals were performed with males only. Both males and females were used for the behavioural analysis of the *Dyrk1a* rescue experiment in Dp1Yey glutamatergic neurons. Due to the difficulty to obtain Dp1Yey; *Dyrk1a*[C/+] mice (three alleles combination and subfertility of the Dp1Yey line with only around 30% transmission of the Dp1Yey allele) and according to the fact that no sex effect was observed in behavior tests done another trisomic model [105], both males and females were pooled for statistical analysis (males and females are represented with different colours on the graphs). Number of animals and age of the animals for each test are given in S10 Table. Protocols for the different tests are described in S1 Materials and Methods. For all these tests, mice were kept in ventilated cages with free access to food and water. The light cycle was controlled as 12 h light and 12 h dark (lights on at 7:00 AM) and the tests were conducted between 8:00 AM and 4:00 PM. Animals were transferred to the experimental room 30 min before each experimental test. Behavioural experimenters were blinded as to the genetic status of the animals. All the standard operating procedures for breeding and behavioural phenotyping have been already described [47,106,107] and are detailed in S1 Materials and Methods.

## Statistical analysis

Statistical analyses were performed using SigmaPlot software. For histological assessments and behavioral tests comparing *Dyrk1a*[Camk2aCre/Camk2aCre] animals to controls, statistical analyses were performed using unpaired t-test when appropriate or the non-parametric Mann-Whitney rank sum test unless otherwise stated in the text. For the four groups analyses (Dp1Yey, Dp1Yey, Dp1Yey; *Dyrk1a*[C/+], Dp1Yey, *Dyrk1a*[C/+] and controls) a two-way ANOVA did not

reveal a significant effect of sex and no interaction with the genotype. Therefore, the sex factor was dropped from the model and a one-way ANOVA and *post hoc* Tukey's multiple comparison test were used to analyse differences between the four genotype groups.

## Supporting information

**S1 Materials and Methods. Supplementary Materials and Methods.**
(DOCX)

**S1 Table. List of Up and Down regulated genes in *Dyrk1a*^C/C^ hippocampi compared with controls (Deseq algorithm, P<0.025).**
(XLSX)

**S2 Table. List of enriched cell populations in deregulated *Dyrk1a*^C/C^ hippocampal genes.**
(XLSX)

**S3 Table. List of the biological processes that are enriched when looking at the list of deregulated oligodendrocyte markers (see the list of the genes in S2 Table) using FunRich (http://www.funrich.org/).**
(XLSX)

**S4 Table. List of early and late response genes deregulated in *Dyrk1a*^C/C^ hippocampus.**
(XLSX)

**S5 Table. List of proteins that are impacted by the different genetic conditions in the proteomic analysis.**
(XLSX)

**S6 Table. List of GO and pathways enriched in the proteome analysis.**
(XLSX)

**S7 Table. List of mouse lines with their genetic background.**
(XLSX)

**S8 Table. List of primers and probes used for genotyping and QRT-PCR analysis.**
(XLSX)

**S9 Table. Information on ddPCR experimental design and data analysis.**
(XLSX)

**S10 Table. List of mice used for behavioral analyses.**
(XLSX)

**S1 Fig. DYRK1A (red) co-localizes with CAMK2A expressing neurons (green) in the glutamatergic pyramidal neurons of the CA1-3, the granular neurons of the dentate gyrus (DG) and in the cortex (photo at the level of the auditory cortex) of a 3-month-old wild-type control mouse.**
(TIF)

**S2 Fig. Histomorphological analysis of the hippocampus of *Dyrk1a*^C/C^ mice.** (A) Representative coronal section of hippocampus at Bregma -1.5 stained with cresyl violet and luxol blue that were used for measurements (Magnification 20X) and dot plots indicating the thickness of the different cellular and molecular layers. (B) Enlarged image of the CA1 showing the selected area made for counting the number of cells within the CA1 and dot plots for the area of the CA1, the number of cells within this area and the cell density. Data are presented as point plots with mean ± SD (n = 3 females aged 3 months per genotype). Pyr: pyramidal layer,

Mol: molecular layer, Gran: granular layer, Or: oriens layer, Rad: radiatum layer.
(TIF)

**S3 Fig. Further analysis of behavior and cognition induced by the inactivation of *Dyrk1a* in the *Camk2a* domain.** (A-B) Effects of *Dyrk1a* inactivation on circadian activity. Locomotor activity during circadian analysis was comparable between *Dyrk1a*$^{C/C}$ and control mice through the light/dark cycle. Numbers on the Y-axis represent the hours. Data are presented as mean ± SEM for each hour. (B) Dot plot of the total number of rears registered during the whole 35H-period of circadian analysis. (C) The total distance travelled during 30 min within the OF was comparable between genotypes. (D) Working memory assessed by percentage of spontaneous alternation within the arms of the Y maze was not impacted by inactivation of *Dyrk1a* in *Dyrk1a*$^{C/C}$ mice. (E) The locomotor activity assessed by the number of arm entries was also similar between the two genotypes. (F) In the fear conditioning test, the baseline level of immobility (precue 1 and precue 2) and the cued freezing performances (cue 1 and cue 2) in a new context were comparable between genotypes. (G-I) Assessment of social behavior in the Crawley three-chamber test shows that both genotypes spend more time exploring the cage containing a congener than the empty cage (G; paired t-test congener vs empty cage: ctrl, ***p<0.001 and *Dyrk1a*$^{C/C}$ ***p<0.001) and exploring the novel than familiar congener (H; paired t-test new congener vs familiar congener: ctrl, **p = 0.002 and *Dyrk1a*$^{C/C}$ *p = 0.019). (I) Social contact assessed by measuring to time spent sniffing both congeners during the test for novelty preference was similar between mutant and control mice. A-I: tests were done on males (1.5–3.5 months old depending on the test with animals aged ± 3 weeks), n = 8–10 per genotype. Data are presented with mean±SD. (J-K) Epileptic susceptibility was tested with the injection of two doses of PTZ in 6 months old male mice. Percentage of mice reaching myoclonic, clonic and tonic seizure stage were similar between the two genotypes at dose 30 mg/kg body weight (J; n = 25 ctrl and n = 20 *Dyrk1a*$^{C/C}$ mice) and 50 mg/kg body weight (K; n = 22 ctrl and n = 19 *Dyrk1a*$^{C/C}$ mice).
(TIF)

**S4 Fig. Representative image of the corpus callosum (cc) showing the selected area in which OLIG2+ cells were counted.** A) distance of 1 mm was measured and the underneath corpus callosum was selected. The cc area as well as the number of OLIG2+ cells and the OLIG2+ cell density did not differ between *Dyrk1a*$^{C/C}$ and control animals. Data are presented as point plots with mean ± SD (each dot represents the mean count of 3 serial sections).
(TIF)

**S5 Fig. Complementary behavioural analysis of DYRK1A dose in different mouse genetic models.** (A) The locomotor activity assessed by the number of arm entries in the Y maze was similar between control, Dp1Yey, *Dp1Yey/Dyrk1a*$^{C/+}$ and *Dyrk1a*$^{C/+}$ genotypes. (B-C) Activity (B) and working memory (C) was assessed in *Dyrk1a* full heterozygous knockout (*Dyrk1a*$^{+/-}$) mice in the Y maze showing no effect of *Dyrk1a* haploinsufficiency (only males were analyzed here). (D) In the fear conditioning test, the baseline level of immobility during the habituation period was similar between genotypes and contextual freezing performance in the same environment after conditioning was also comparable between genotypes. (E-F) Assessment of social behavior in the Crawley three-chamber test shows that all genotypes spend more time exploring the cage containing a congener than the empty cage (E; paired t-test congener vs empty cage: ctrl, ***p<0.001; Dp1Yey, *p = 0.03; Dp1Yey/*Dyrk1a*$^{C/C}$, *p = 0.01; *Dyrk1a*$^{C/C}$, p***p<0.001). Moreover, no difference was found between genotypes in the total time spent sniffing the cage containing a congener (F). Data are represented as point plots with mean

±SD. Males are in blue and females are in red.
(TIF)

**S6 Fig. Validation of the full knock-out allele.** (A) Photo of a wild-type (WT) and a *Dyrk1a^-/-* embryos at E10.5 showing growth retardation of the *Dyrk1a* knockout animal. (B) Western blot analysis of equivalent amounts of protein extracts from WT and *Dyrk1a^-/-* embryos showing the absence of the Dyrk1a-specific protein at around 100 kDa. A band is seen at around 55kDa, but this is an unspecific band that appears sometimes in our blots as seen also in wild-type protein extracts in panel E. (C) Photo of a 4-months-old WT male and *Dyrk1a^+/-* littermate showing significant body size reduction. (D) Body weight of WT and *Dyrk1a^+/-* twelve weeks old males. (E) Western blot analysis: autoradiographic image and quantification of immunoblots of Dyrk1a protein in the hippocampus of WT and *Dyrk1a^+/-* animals (8 WT and 7 *Dyrk1a^+/-* 12-weeks old males). Band intensities were estimated using ImageJ and normalized against the loading control Gapdh (or against total loaded proteins visualized with Ponceau red). Data are presented as point plots with mean ± SD with unpaired Mann-Whitney test, $^*p<0.05$, $^{***}p<0.001$.
(TIF)

## Acknowledgments

We would like to thank members of the research group, of the IGBMC laboratory and of the ICS. We are grateful to the IGBMC proteomic plateform and Doulaye Dembele for their expert technical assistance in proteomic analysis, and Binnaz Yalcin and Stephan Collins for their help in brain morphometric analysis. We extend our thanks to the animal care-takers of the ICS who oversee the mice wellness.

## Author Contributions

**Conceptualization:** Véronique Brault, Giovanni Iacono, Yann Hérault.

**Formal analysis:** Véronique Brault, Thu Lan Nguyen, Javier Flores-Gutiérrez, Giovanni Iacono.

**Funding acquisition:** Yann Hérault.

**Investigation:** Véronique Brault, Thu Lan Nguyen, Javier Flores-Gutiérrez, Giovanni Iacono, Marie-Christine Birling, Valérie Lalanne, Hamid Meziane, Antigoni Manousopoulou, Guillaume Pavlovic, Loïc Lindner, Mohammed Selloum, Tania Sorg, Spiros D. Garbis, Yann Hérault.

**Methodology:** Véronique Brault, Giovanni Iacono, Hamid Meziane, Antigoni Manousopoulou, Guillaume Pavlovic, Loïc Lindner, Mohammed Selloum, Spiros D. Garbis, Yann Hérault.

**Project administration:** Yann Hérault.

**Resources:** Marie-Christine Birling, Tania Sorg, Eugene Yu.

**Supervision:** Mohammed Selloum, Tania Sorg, Yann Hérault.

**Validation:** Véronique Brault, Thu Lan Nguyen, Giovanni Iacono, Spiros D. Garbis, Yann Hérault.

**Visualization:** Véronique Brault, Giovanni Iacono, Yann Hérault.

**Writing – original draft:** Véronique Brault, Thu Lan Nguyen, Giovanni Iacono, Antigoni Manousopoulou, Guillaume Pavlovic, Loïc Lindner, Mohammed Selloum, Tania Sorg, Eugene Yu, Spiros D. Garbis, Yann Hérault.

**Writing – review & editing:** Véronique Brault, Giovanni Iacono, Yann Hérault.

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
