## [Decision Letter · Decision Letter 0]

29 May 2021

Dear Yann,

Thank you very much for submitting your Research Article entitled 'Dyrk1a gene dosage in glutamatergic neurons has key effects in cognitive deficits observed in mouse models of MRD7 and Down syndrome' to PLOS Genetics.

The manuscript was fully evaluated at the editorial level and by four independent peer reviewers who have expertise in this area. All the reviewers appreciated the attention to an important problem and novelty of this work, but raised some substantial concerns about the current manuscript. As you can see from the reviews, the reviewers requested details and clarifications on experimental methodology and clarification on molecular and anatomical findings (Reviewers 1 and 3), as well as sought careful integration of the multiple facets of the details into a coherent story (Reviewer 2 and 4). Based on the reviews, we will not be able to accept this version of the manuscript, but we would be willing to review a much-revised version. We cannot, of course, promise publication at that time.

If you decide to revise the manuscript for further consideration at PLOS Genetics, please aim to resubmit within the next 60 days, unless it will take extra time to address the concerns of the reviewers, in which case we would appreciate an expected resubmission date by email to plosgenetics@plos.org.

[LINK]

We are sorry that we cannot be more positive about your manuscript at this stage. Please do not hesitate to contact us if you have any concerns or questions.

Yours sincerely,

Santhosh Girirajan

Associate Editor

PLOS Genetics

Gregory Barsh

Editor-in-Chief

PLOS Genetics

Reviewer's Responses to Questions

**Comments to the Authors:**

Reviewer #1: The manuscript by Brault et al., “Dyrk1a gene dosage in glutamatergic neurons has key effects in cognitive deficits observed in mouse models of MRD7 and Down syndrome” is a state of the art study to understand the specific effects of Dyrk1a dosage imbalance in the hippocampus with regards to cognitive deficits. This study examines mouse models with 0, 1, 2, and 3 copies of Dyrk1a specific to the glutamatergic neurons of the hippocampus and cortex, and how the dosage imbalance of Dyrk1a affects learning and memory. These data are important to understanding how DYRK1A dosage imbalance causes phenotypes associated with MRD7 (reduced dosage of DYRK1A) and Down syndrome (increased dosage of DYRK1A). There are a number of issues, however, with the study and methodology that need to be clarified so the results may be analyzed in the proper context of the materials and methods used.

Major concerns:

1. What is the genetic background of the Dyrk1aC/C, Dyrk1a+/-, Dyrk1aCamk2aCre/+, Dp1Yey, Dp1Yey/Dyrk1aCamk2aCre/+, Dyrk1acKO/+, and transgenic Tg(Camk2-Cre)4Gsc mice? Are Dyrk1aCamk2aCre/+, Dp1Yey, and Dp1Yey/Dyrk1aCamk2aCre/+, and Dyrk1acKO/+ all littermates with a similar background (Results beginning in line 294?) What were the breeding schemes used to generate these mice? Is it valid to compare Dyrk1aC/C and Dyrk1aC/+ mice if they are not on the same genetic background (lines 341-342)? Likewise, is it valid to compare results from Dyrk1aC/C and Dyrk1a+/- mice if they are not on the same genetic background (lines 401-407)?

2. Why is it necessary to have a full knockout of Dyr1ka in the glutamatergic neurons to see the cognitive phenotypes whereas humans with a single mutation in DYRK1A show these phenotypes?

3. Please outline the numbers of male and female mice in each experiment and whether the tissues tested came from male or female mice. This would be useful information to have in the materials and methods or all figure legends or both. In Figure 4 and supplementary Figure 4, the mice are colored by sex and this would be helpful in all figures. It was noted, by looking at the separation of colors, especially in 4E of the text and 4E of the supplementary figures, that there could have been some sex-specific differences in NOR retention and social recognition, respectively. Are there enough mice in each experiment to accurately test for sex-specific differences in each phenotype (Lines 759 and 773-774 and 897-898)? Not properly analyzing potential differences between sexes in each experiment is a significant shortcoming of the manuscript.

4. Please state the age of mice used in each experiment. Why were each of these ages chosen? For example, why was the transcriptional profiling done at P30? What was the age of the mice for the morphometric analyses? Is it valid to compare mice that are between 2.5 and 7 months of age on behavioral tests (lines 754 and 755)?

5. In Figure 1A, please show the regions that the inset pictures came from on the hippocampus and cortex. In the hippocampus in Figure 1A, it appears that both DYRK1A and CAMK2A are expressed outside or around the neurons and not in the neurons as compared to the cortex. Please comment on this.

6. The interpretation of morphometric and cellular changes is severely hampered by the methods used to quantify these parameters. In the manuscript these results are reported on a chosen area or cells counted within a chosen area. Stereological methodology should be used to calculate volumes and total numbers of cells in structures, as is the standard. The manuscript should compare volumes of the parts of the hippocampus and total cell number within the hippocampus. In line 170, the manuscript says that “the total number of neurons is unchanged.” This is incorrect. The total number of neurons was not quantified, only a density of neurons in an area that was not detailed as to how it was chosen. How was a particular area (counting frame) chosen for cell counting or other quantification? If the number of neurons were not counted, how can the statement in lines 260-264 be made? Numbers of neurons were not counted, only density. This is a significant deficiency in the current manuscript and needs to be addressed or the present methodology better described and justified.

Minor concerns:

1. The statement “Down syndrome (DS; Trisomy 21), is the first genetic cause of mental retardation” (Lines 71 and 72) is confusing. Perhaps another word in place of “first” such as “major” or “leading?”

2. The concept of a “Down syndrome Critical Region” (lines 73-74) has been disproven many times. It is not “a critical region…associated with DS features including mental retardation.” Please remove this outdated concept from the manuscript.

3. Please use similar axes for similar tests. It would be useful to have Figures 3I and 3J on the same axes for comparison.

4. Some of the references in the manuscript are not defined and still have notes from the authors in them (Lines 292, 316, 429, 434, 458, 469, 483, 491, for example).

5. Person first terminology should be used and the term “patients” should be avoided (line 503). Perhaps “Individuals with DS?”

Reviewer #2: In this study, Brault et al. use conditional knockout technology to study the role of Dyrk1a gene dosage in the function of excitatory glutamatergic neurons in cognition. Using Camk2a-Cre to delete floxed Dyrk1a alleles in glutamatergic neurons of wild type mice or mice modeling trisomy 21, they examine locomotor activity, working and long-term memory, and susceptibility to epileptogenesis. They further examine effects of gene dosage on gene expression. The main conclusion is that Dyrk1a dosage impacts long term memory, possibly through modulation of long-trem synaptic plasticity. The role of the gene in glutamatergic neurons has not been investigated previously, though recent studies have suggested it might be important in synaptic properties of these neurons.

Dyrk1A excess gene dosage is implicated in neurological deficits in Down Syndrome, and loss of this gene is associated with a severe autism syndrome. Because the gene is involved in multiple processes in brain development and function, it is difficult to pinpoint specific loci of its action via conventional gene knockout. This study adds to our knowledge concerning the postnatal actions of this gene. There are a number of areas in which it is difficult to draw a clear relationship between the effects of gene dosage, the anatomical substrates, and the alterations in gene expression, and the clarity of the study would be improved by less extrapolation of the data and speculation regarding these relationships.

1. Page 71 DS is “the first identified genetic cause of mental retardation”

2. Results line 142 and following: did the authors investigate in detail the degree of specificity of Camk2a driven deletion for glutamatergic neurons?

3. Line 160-how might the changes in cell morphology or tissue organization in the cortex have affected the readouts of behavioral tests? What neurons (or other cell types) were affected? What behavioral changes may be said with confidence to relate solely to deletion in hippocampal neurons?

4. Line 251 and following: scRNA-seq might have been quite informative here.

5. Line 256 following: what exactly is going on with the oligodendrocytes

6. Line 273 these genes were upregulated in what cell type? The authors might at least perform immunostaining or RNA-Scope for the more important ones?

7. Line 316-remove “Duchon and Herault HMG?

8. Did the authors do proteomics in Dyrk1aC/C mice to compare with RNA-seq?

9. Line 391-were blocking peptides used to check specificity of pulldown?

10. Line 397 and following: the Discussion is far too long; statements that are largely speculative and center around the many remaining unknowns could be removed or truncated to help the reader focus on the strong conclusions of the study. For example: role of IEG genes, normalization of gene copy number and memory, LTP.

11. Line 434-insert numbered reference Duchon et al.

12. Line 573 following: clarify the genetic backgrounds of the various mouse mutants actually studied.

Reviewer #3: Review attached

Reviewer #4: This manuscript by Brault et al. reports findings from manipulating Dyrk1a gene dosage in glutamatergic neurons. Homozygous and heterozygous conditional mutants for Dyrk1a using CamK2-Cre, and this was combined with Dp1Yey trisomy model. Effects of Dyrk1a deletion in glutamatergic neurons on brain morphology and behavior were assessed, revealing deficits in a variety of behavioral domains. Behavioral consequences of reducing Dyrk1a dosage in a Dp1Yey background was also tested. Proteomic analysis of differentially abundant proteins in mutant backgrounds identified synaptic proteins that are candidate interactors with Dyrk1a. Overall, this manuscript reports multiple interesting molecular, anatomical and behavioral effets of CamK2-Cre-mediated deletion of Dyrk1a. However, these observations are not tied together into a coherent story and there are several areas where the manuscript could be improved, listed below.

MAJOR CRITIQUES

- A major weakness of this study is that there are no analyses at the level of neuronal structure or function that could connect the observations of abnormal abundance of key neuronal proteins, altered brain morphology and density of cortical neurons with abnormal behavior in conditional Dyrk1a mutants. This is a major gap in the study design. For example, a direct measure of glutamatergic neuronal morphology would corroborate the interesting finding of altered density.

- The logic of using CamK2-Cre to delete Dyrk1a should be better explained. Why was this line, which will delete Dyrk1a later in the development of glutamatergic neurons, used? There may be a compelling reason, e.g. circumventing early developmental functions of Dyrk1a, but this should be explained.

- Relatedly, another study (https://doi.org/10.1016/j.biopsych.2021.01.012) has recently been published that deleted Dyrk1a using Emx1-Cre, resulting in an earlier developmental loss of Dyrk1a function in glutamatergic neurons. While it is understandable that this paper was not cited given the timing of publication, the findings of the two studies (i.e. earlier and later deletion of Dyrk1a in glutamatergic neurons) are complementary and should be discussed.

- The description of the RNAseq experiment could be written more clearly and succinctly.

- For reporting of behavioral results, males and females are pooled and represented with different colored dots in Figure 4, while this is not done in Figure 3. Why are the results reported differently between these two figures? Why where sexes pooled instead of analyzed separately?

MINOR CRITIQUES

- Lines 103-105 require a reference.

- The amount of hippocampal GAPDH in figure 1H appears different between control and mutant. Is this due to different total protein concentration between the two samples? Or is there actually a change in GAPDH levels in mutants?

**Have all data underlying the figures and results presented in the manuscript been provided?**

Reviewer #1: Yes

Reviewer #2: Yes

Reviewer #3: Yes

Reviewer #4: Yes

PLOS authors have the option to publish the peer review history of their article (what does this mean?). If published, this will include your full peer review and any attached files.

Reviewer #1: No

Reviewer #2: No

Reviewer #3: No

Reviewer #4: No

---

## [Decision Letter · Decision Letter 1]

16 Aug 2021

Dear Dr Herault,

We are pleased to inform you that your manuscript entitled "Dyrk1a gene dosage in glutamatergic neurons has key effects in cognitive deficits observed in mouse models of MRD7 and Down syndrome" has been editorially accepted for publication in PLOS Genetics. Congratulations! However, we do ask the authors to consider the additional comments provided by Reviewer 3, including correct any errors, references, and clarify statements related to data interpretation in the final version manuscript. 

Yours sincerely,

Santhosh Girirajan

Associate Editor

PLOS Genetics

Gregory Barsh

Editor-in-Chief

PLOS Genetics

Comments from the reviewers (if applicable):

Reviewer's Responses to Questions

**Comments to the Authors:**

Reviewer #1: The authors have addressed my questions in the revised manuscript.

Reviewer #2: The authors have provided satisfactory responses to my queries, and have amended the text where appropriate.

Reviewer #3: The authors have satisfactory addressed most of the questions regarding the experimental methodology and clarified some aspects of the mice used in the study, which have substantially improved the manuscript.

As requested, they also added new immunostainings showing that neuronal and glial cell densities are affected in the SSC of Dyrk1aC/C mice (new figure 3) but the explanation of this finding still missing. Why knocking-down DYRK1A in postnatal glutamatergic neurons affects cell densities? The authors need to comment on this in the context of the other experiments performed in this mutant model and provide morphological data. In the discussion they wrote “…..with Dyrk1aC/C brain revealing the impact of DYRK1A on postnatal neuronal morphogenesis.” This is an overstatement. The authors cannot correlate cell densities with defects in neuronal morphogenesis or remodelling without providing any experimental data. The phenotype of cortical pyramidal neurons reported in other Dyrk1a mutant mice cannot be directly extrapolated to Dyrk1aC/C mutants. Likewise, the fact that alterations in the dosage of DYRK1A affect neurite outgrowth in differentiating neurons in culture, as shown in Dang et al. 2016, do not necessary mean that DYRK1A is necessary for shaping the morphology of cortical neurons during late postnatal development.

Another weakness of the manuscript that has not been addressed is the significance of the RNA-seq results. The authors should perform additional experiments to show by other methods the dysregulation of a selected set of genes (immediate-early genes as suggested by other reviewer and/or some of the down-regulated oligodendroglial genes, Sox8 for instance). These experiments are doable and do not require a large number of animals. The possibility that depletion of DYRK1A in glutamatergic neurons could affect the expression of oligodendroglial genes is interesting but the data is too short to reach any conclusion. This part of the results is much better explained in the new version of the manuscript but I still found the text too long and difficult to follow. Regarding the age of the animals used in the experiment, postnatal day 30. The sentence in line 268 is not correct. The mouse brain is considered mature at age two months although some of the developmental features, for instance reduction of neocortical thickness, continuous after this age (see Hammelrath et al. NeuroImage 2016). In my opinion, this do not diminish the potential interest of the results but it has to be considered for their proper interpretation.

There are some mistake, omissions and misleading sentences in the manuscript:

- Sentences in lines 85–89 requires references.

- Reference 16 in line 91 is not correct.

- As the mouse brain is considered fully developed at 2-3 months, far beyond the onset of Cre expression in TgCamK2aCre brains, I suggest to add “early” in the sentence in line 138, ”…Dyrk1a on “early” brain development.”

- The sentence in lines 173-174 “Dyrk1a was reported to impact cortical thickness especially at the level of the somatosensory cortex, with no impact on overall cell density (34)” is not correct. Guedj et al. (ref 34) show differences in cell densities of different brain regions, including the SSC, in adult Dyrk1a+/- and tgDyrk1a mice.

- In line 431, the word “mice” is missing at the end of the sentence. References 19 and 54 in the same sentence (line 433) are not correct.

Other suggestions:

- For clarity, I suggest to indicate in the Abstract that mutations in DYRK1A gene are “loss-of function” mutations (lines 36-37) and add “postnatal” before “glutamatergic neurons..” in line 43.

- Also for clarity indicate in the Introduction that mutations in DYRK1A (line 78) are “loss-of-function” mutations. The paragraph “in most cases…..” (lines 83-85) can be eliminate because most missense DYRK1A variants seem to affect the enzymatic activity (see Arranz et al. Neurobiol Dis, 2019) and therefore both truncating and missense mutations are considered loss-of-function mutations.

- Numbers in figure 1 (panel E) appear fuzzy and are difficult to read.

- For clarity indicate in the histograms of figure 3 the cell type (total cells, NeuN+ or S100b+ cells) in the Y axis.

- I suggest to use reviews to shorten the reference list.

Reviewer #4: The authors have adequately addressed critiques raised in my previous review. No further comments.

**Have all data underlying the figures and results presented in the manuscript been provided?**

Reviewer #1: Yes

Reviewer #2: Yes

Reviewer #3: Yes

Reviewer #4: Yes

PLOS authors have the option to publish the peer review history of their article (what does this mean?). If published, this will include your full peer review and any attached files.

Reviewer #1: No

Reviewer #2: No

Reviewer #3: No

Reviewer #4: No

**Data Deposition**

http://datadryad.org/submit?journalID=pgenetics&manu=PGENETICS-D-21-00559R1

**Press Queries**

---

## [Editor Report · Acceptance letter]

15 Sep 2021

PGENETICS-D-21-00559R1 

*Dyrk1a* gene dosage in glutamatergic neurons has key effects in cognitive deficits observed in mouse models of MRD7 and Down syndrome 

Dear Dr Herault, 

We are pleased to inform you that your manuscript entitled "*Dyrk1a* gene dosage in glutamatergic neurons has key effects in cognitive deficits observed in mouse models of MRD7 and Down syndrome" has been formally accepted for publication in PLOS Genetics! Your manuscript is now with our production department and you will be notified of the publication date in due course.

With kind regards,

Andrea Szabo

PLOS Genetics

On behalf of:
